# Maintenance of type 2 glycolytic myofibers with age by Mib1-Actn3 axis

Ji-Yun Seo [1], Jong-Seol Kang[1], Ye Lynne Kim [1], Young-Woo Jo [1], Ji-Hoon Kim [1], Sang-Hyeon Hann[1], Jieon Park [1], Inkuk Park [1], Hyerim Park [1], Kyusang Yoo [1], Joonwoo Rhee [1], Jung-Wee Park [2], Yong Chan Ha[2] & Young-Yun Kong [1]✉

Age-associated muscle atrophy is a debilitating condition associated with loss of muscle mass and function with age that contributes to limitation of mobility and locomotion. However, the underlying mechanisms of how intrinsic muscle changes with age are largely unknown. Here we report that, with age, Mind bomb-1 (Mib1) plays important role in skeletal muscle maintenance via proteasomal degradation-dependent regulation of α-actinin 3 (Actn3). The disruption of *Mib1* in myofibers (Mib1$^{\Delta MF}$) results in alteration of type 2 glycolytic myofibers, muscle atrophy, impaired muscle function, and Actn3 accumulation. After chronic exercise, Mib1$^{\Delta MF}$ mice show muscle atrophy even at young age. However, when Actn3 level is downregulated, chronic exercise-induced muscle atrophy is ameliorated. Importantly, the Mib1 and Actn3 levels show clinical relevance in human skeletal muscles accompanied by decrease in skeletal muscle function with age. Together, these findings reveal the significance of the Mib1-Actn3 axis in skeletal muscle maintenance with age and suggest the therapeutic potential for the treatment or amelioration of age-related muscle atrophy.

---

[1] School of Biological Sciences, Seoul National University, Seoul, South Korea. [2] Department of Orthopaedic Surgery, Chung-Ang University College of Medicine, Seoul, South Korea. ✉email: ykong@snu.ac.kr

Age-related muscle atrophy is associated with a decrease in muscle mass, impaired muscle function, and selective loss of type 2 glycolytic myofibers or reduction of its cross-sectional area (CSA), which leads to increases in frailty and mortality and loss of autonomy in the elderly[1]. Since life expectancy has increased dramatically, age-associated muscle atrophy became a serious social and medical concern[2]. Therefore, a better understanding of molecular mechanisms underlying the development of age-associated muscle atrophy is of utmost importance to prevent and treat age-associated muscle atrophy.

Besides the loss of muscle mass, another feature of age-associated muscle atrophy is the deterioration in muscle quality due to alteration of muscle structural composition[3], loss of proteostasis[4], and accumulation of sarcomeric proteins[5]. Myofibrillar accumulation and/or sarcomeric protein aggregates are detrimental to tissue homeostasis and maintenance due to constant proteotoxic stress in aged skeletal muscle[4,6]. Hence, appropriate protein degradation and protein turnover are instrumental for skeletal muscle maintenance throughout the lifetime. Among several systems responsible for proteostasis, the ubiquitin–proteasome system (UPS) is one of the tightly coordinated systems responsible for the removal of aggregated and dysfunctional proteins[7,8]. Dysfunctional UPS and accumulation of ubiquitinated and/or aggregated proteins result in pathological responses in skeletal muscles[9,10]. The to-be-degraded proteins are highly regulated by E3 ubiquitin ligase, which recognizes the substrates and adds ubiquitin moieties to target proteins, thereby triggering the degradation of ubiquitin-conjugated substrates[8]. Several E3 ubiquitin ligases, such as muscle RING finger 1 (MuRF1), atrogin-1/MafBx, and tripartite motif-containing 32 (Trim32), are known to play a critical role in muscle atrophy, such as disuse, disease, fasting, and denervation[11]. However, it is still controversial whether they play significant roles in aged skeletal muscle[12,13]. Thus, further studies are necessary to elucidate which E3 ubiquitin ligase is implicated in skeletal muscle maintenance and function with age.

Actn3, an actin-binding protein, is one of the major components of skeletal muscle Z-disk in type 2 glycolytic myofibers[14]. To date, Actn3 is known as "a gene of speed" because the Actn3 genotype is closely associated with skeletal muscle performance[15]. Humans or mice with loss of Actn3 showed increased endurance function and reduced grip strength[16,17]. Inversely, the overexpression of Actn3 in skeletal muscle can be detrimental to skeletal muscle[18], suggesting a dosage effect of Actn3 on skeletal muscle. Since Actn3 interacts with a broad range of proteins involved in structural, metabolic, and signaling proteins, as well as calcium handling[19], it is critical to retain balanced levels of Actn3 to maintain skeletal muscle and sarcomere integrity. Since Z-disk is a hub for mechanosignaling, several E3 ubiquitin ligases are prone to target Z-disk proteins[20]. Actn3 is known to interact with and be ubiquitinated by Trim32 during atrophic condition such as denervation[21]. However, it is still unknown how Actn3 is regulated in skeletal muscle with age.

Mindbomb-1 (Mib1) is an E3 ubiquitin ligase that regulates Notch signaling[22,23]. Thus, Mib1 has been of great interest in understanding how Notch signaling is implicated in development, tissue maintenance, and cell differentiation[24]. Apart from Notch signaling, recent studies have shown putative substrates of Mib1 in various biological processes[25]. Although recent studies showed the Notch-independent role of Mib1 in other tissues[26–28], it is still unknown whether Mib1 plays a protein degradation-dependent role in skeletal muscle.

Here, we investigated the importance of Mib1 in muscle maintenance and its role in the regulation of type 2 glycolytic myofibers with age. We showed that, with age, sustained loss of Mib1 in myofibers leads to muscle atrophy, impaired muscle function, and abnormal histological features. These muscle atrophic features of myofiber-specific Mib1-deficient (MCK-Cre; Mib1$^{f/f}$, hereafter Mib1$^{\Delta MF}$) mice were observed after chronic moderate exercise at a young age, which was ameliorated upon downregulation of Actn3 in skeletal muscle. Our data thus demonstrate a novel role of Mib1 in skeletal muscle maintenance of type 2 glycolytic myofibers with age by regulating Actn3. In addition, we further suggest that the maintenance of type 2 glycolytic myofibers via the Mib1–Actn3 axis might serve well as a therapeutic target for age-associated muscle atrophy and will broaden our understanding of the disease.

## Results

**Muscle atrophy in Mib1$^{\Delta MF}$ mice with age.** Since several E3 ubiquitin ligases are implicated in aging[7], we examined the age-related changes in Mib1 expression in skeletal muscle. We observed that Mib1 expression was significantly decreased in aged skeletal muscle (Fig. 1a, b). To determine whether the loss of Mib1 with age is implicated in age-associated muscle atrophy, we generated Mib1$^{\Delta MF}$ mice and analyzed them with age. The body weights of Mib1$^{WT}$ and Mib1$^{\Delta MF}$ mice were comparable, but the masses of hindlimb muscles (tibialis anterior (TA), gastrocnemius (GA), quadricep (Q)) from 16-month-old Mib1$^{\Delta MF}$ mice were significantly decreased (Fig. 1c–g). In addition, 16-month-old Mib1$^{\Delta MF}$ mice showed significant increases in fat masses (inguinal (Ing) fat, epididymal fat, (Epi) and visceral (Vis) fat) compared to those of Mib1$^{WT}$ mice (Fig. 1h–k). Taken together, these results indicate the age-associated muscle atrophy in Mib1$^{\Delta MF}$ mice.

In Mib1$^{\Delta MF}$ mice, muscle stem cells are depleted at 3 months of age because juvenile cycling muscle stem cells (MuSCs) cannot be converted into adult quiescent MuSCs due to defective Notch signaling[29,30]. Since either depletion of MuSCs or impaired Notch signaling can lead to age-associated muscle atrophy, we analyzed MuSC-ablated mice (Pax7-CreERT;Rosa-DTA (diphtheria toxin A), hereafter SC$^{DTA}$) and Notch-deficient mice (Pax7-CreERT;Rbpjk$^{f/f}$ and Pax7-CreERT;Notch1$^{f/f}$;Notch2$^{f/f}$, hereafter Notch$^{\Delta SC}$). Consistent with the previous study[31], the muscle masses of 16-month-old SC$^{DTA}$ mice were comparable to those of control mice (Supplementary Fig. 1a–e). Moreover, 16-month-old Notch$^{\Delta SC}$ mice did not show muscle atrophy (Supplementary Fig. 2a–d). Taken together, these results showed that the sustained loss of Mib1 in myofiber leads to age-associated muscle atrophy irrespective of MuSCs and Notch signaling.

**Selective alteration of type 2 glycolytic myofibers in Mib1$^{\Delta MF}$ mice.** To investigate whether age-associated muscle atrophy of Mib1$^{\Delta MF}$ mice is due to the alteration in size or loss of myofiber numbers, the hindlimb muscles of 16-month-old Mib1$^{WT}$ and Mib1$^{\Delta MF}$ mice were immunohistochemically analyzed. Sixteen-month-old Mib1$^{\Delta MF}$ hindlimb muscles showed a decrease in CSA and whole myofiber numbers (Fig. 2a–d). Interestingly, CSA or numbers of type 2 glycolytic myofibers were decreased compared to those of Mib1$^{WT}$ hindlimb muscles, while those of oxidative fibers were comparable (Fig. 2e–g). These results indicate that similar to age-associated muscle atrophy, the selective decrease in CSA or number of type 2 glycolytic myofibers in hindlimb muscle leads to loss of muscle mass in 16-month-old Mib1$^{\Delta MF}$ hindlimb muscle. Together, these results suggest that with age, Mib1 might be essential to maintain type 2 glycolytic myofibers and ultimately skeletal muscle.

**Age-associated abnormal skeletal muscle characteristics in Mib1$^{\Delta MF}$ mice.** Contrary to the onset of age-associated muscle atrophy, which usually occurs at ~24 months of age[31], Mib1$^{\Delta MF}$ mice showed a significant reduction in muscle mass with age,

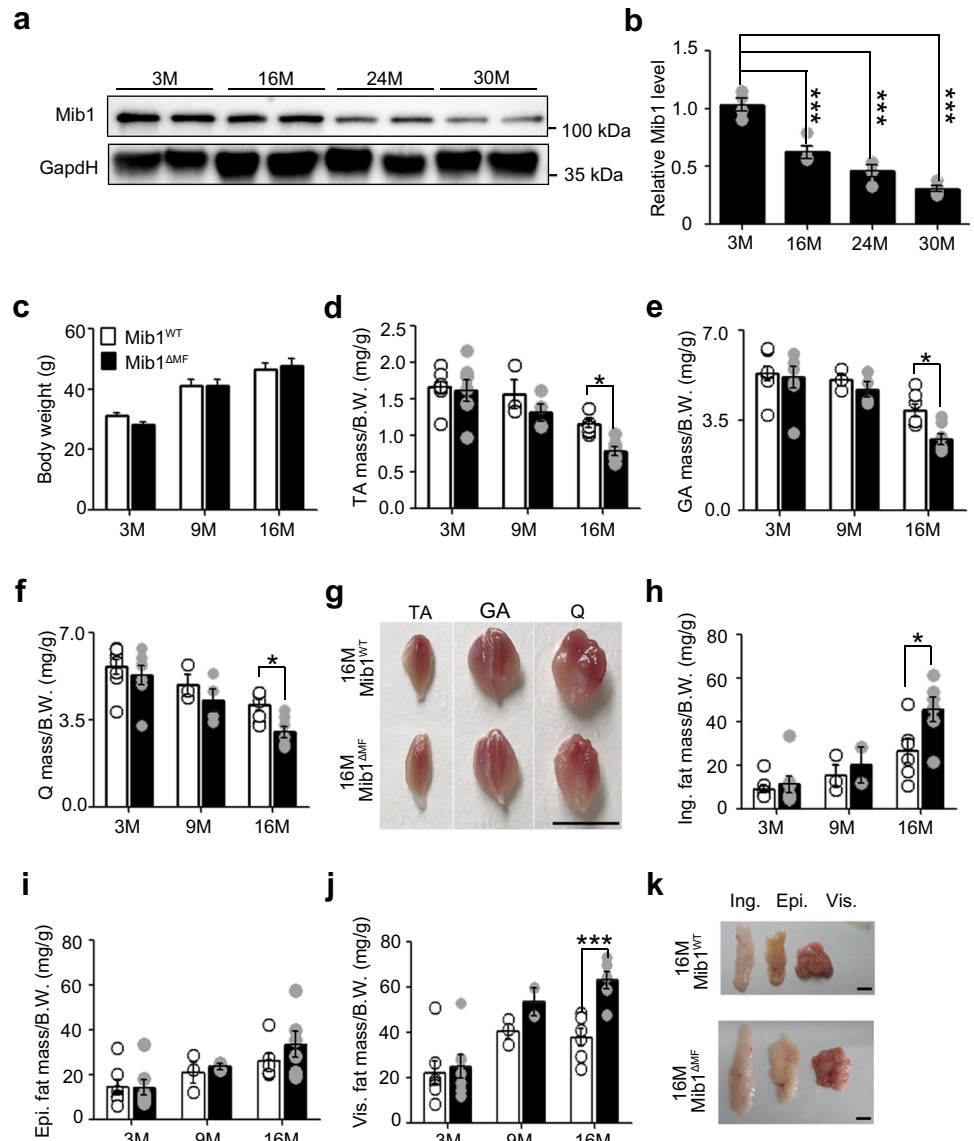

**Fig. 1 Age-associated muscle atrophy in Mib1$^{\Delta MF}$ mice. a**, **b** Immunoblotting (IB) of Mindbomb-1 (Mib1) from wild-type (WT) gastrocnemius (GA) muscles at indicated ages (**a**) and quantification of western blot analysis (**b**). The intensity of Mib1 expression at indicated ages (M; month) was quantified by densitometry. **c**–**f** Body weights (**c**) and relative tibialis anterior (TA) (**d**), GA (**e**), and quadricep (Q) (**f**) muscle masses to body weights of *Mib1$^{f/f}$* (Mib1$^{WT}$) and *MCK-Cre;Mib1$^{f/f}$* (Mib1$^{\Delta MF}$) at indicated ages. **g** Gross morphology of hindlimb muscles (TA, GA, and Q muscles) from 16-month-old Mib1$^{WT}$ and Mib1$^{\Delta MF}$ mice. Scale bars, 0.5 cm. **h**–**j** Inguinal (Ing) fat (**h**), epididymal (Epi) fat (**i**), and visceral (Vis) fat (**j**) masses to body weights of Mib1$^{WT}$ and Mib1$^{\Delta MF}$ at indicated ages. **k** Gross morphology of fat (Ing, Epi, and Vis fats) from 16-month-old Mib1$^{WT}$ and Mib1$^{\Delta MF}$ mice. Scale bars, 1 cm. Data are presented as means ± s.e.m. Data are shown as representatives of at least three independent experiments. *n* = 4 (**b**), *n* = 11, 18, and 17 for 3-, 9-, and 16-month-old Mib1$^{WT}$ mice and 16, 18, and 16 for 3-, 9-, and 16-month-old Mib1$^{\Delta MF}$ mice, respectively (**c**); *n* = 8, 3, and 7 for 3-, 9-, and 16-month-old Mib1$^{WT}$ mice and 7 for 3-, 9-, and 16-month-old Mib1$^{\Delta MF}$ mice, respectively (**d**–**f**, **h**–**j**); *n* = 7, 3, and 6 for 3-, 9-, and 16-month-old Mib1$^{WT}$ mice and *n* = 7, 4, and 6 for 3-, 9-, and 16-month-old Mib1$^{\Delta MF}$ mice, respectively (**d**–**f**, **h**–**j**). One-way ANOVA for (**b**). Two-way ANOVA for (**c**–**f**, **h**–**j**). *$P < 0.05$; ***$p < 0.001$.

leading us to assume the acceleration of age-associated muscle atrophy in Mib1$^{\Delta MF}$ mice. Thus, we further examine the histopathological changes related to age-associated muscle atrophy in skeletal muscle. Histological analysis of GA muscle in 16-month-old Mib1$^{\Delta MF}$ mice showed increased vacuolated fibers, a common histopathological feature in degenerating muscle in aging[32,33] (Fig. 2h, i). The ultrastructure of Q muscles was assayed by transmission electron microscopy (TEM). Sixteen-month-old Mib1$^{WT}$ Q muscle showed uniformed and evenly spaced sarcomeres (Fig. 2j, k), while Mib1$^{\Delta MF}$ Q muscle exhibited abnormal sarcomeric structures such as abnormal accumulation

of tubular aggregates (Fig. 2l, n; asterisk mark), Z-disk misalignment (Fig. 2m; arrow), aberrant membrane structure (Fig. 2n; arrow), and irregularly shaped and dilated sarcoplasmic reticulum tubules, which might be a precursor form of tubular aggregates (Fig. 2o; arrow). The notable ultrastructure abnormalities in age-associated muscle atrophy are large tubular aggregation, an ordered aggregates of elongated sarcoplasmic reticulum tubules and known to appear ~24 months of age, and disorganized sarcomere structure[33–35]. These abnormal structures observed in 16-month-old Mib1$^{\Delta MF}$ mice consolidated the aging-like phenotypes. Taken together, these histopathological changes

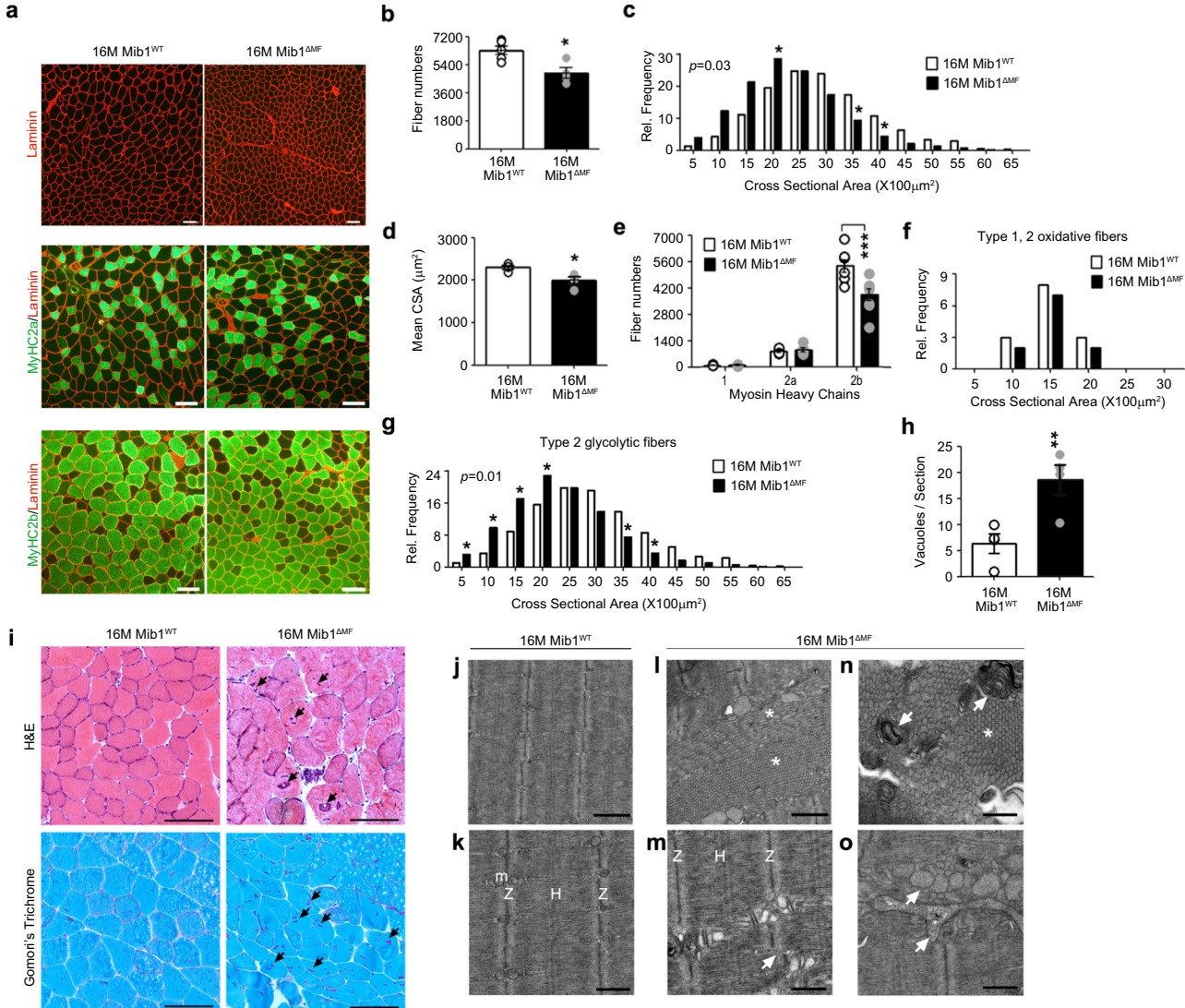

**Fig. 2 Selective alteration of type 2 glycolytic myofiber and abnormal skeletal muscle characteristics in Mib1$^{\Delta MF}$ mice.** Cross-sections of 16-month-old Mib1$^{WT}$ and Mib1$^{\Delta MF}$ hindlimb muscles (TA (**a**, **c**, **d**, **f**, **g**) and GA (**b**, **e**) muscles) were subjected to immunohistochemistry (IHC) staining and analyzed. **a** Representative images of IHC staining for MyHC2a (green; type 2 oxidative myofibers), MyHC2b (green; type 2 glycolytic myofibers), and laminin (red). **b** Quantification of whole myofiber numbers ($p = 0.014$). **c**, **d** Morphometric quantification of cross-sectional area (CSA) (**c**) and mean CSA (**d**) of whole myofibers ($p = 0.0214$ for **d**). **e** Quantification of myofiber numbers by fiber types. MyHC1 (type 1 oxidative myofibers), MyHC2a, MyHC2b indicate type 1, type 2 oxidative (2a), and type 2 glycolytic (2b) myofibers, respectively. **f**, **g** Morphometric quantification of CSA of type 1 and 2 oxidative myofibers (**f**) and type 2 glycolytic myofibers (**g**). **h**, **i** Hematoxylin and eosin (upper panel) and Gomori's trichrome (lower panel) of GA muscles (**i**) and quantification of vacuoles per sections (**h**) ($p = 0.0058$ for **h**). Arrows indicate vacuoles. **j–o** Ultrastructural analysis of Q muscles of 16-month-old Mib1$^{WT}$ (**j**, **k**) and Mib1$^{\Delta MF}$ (**l–o**). m, Z, and H indicate mitochondria, Z-disk, and H-band, respectively. Arrows indicate Z-disk misalignment for **m**, abnormal membrane structure for **n**, and irregularly shaped and dilated sarcoplasmic reticulum (SR) tubules for **o**. Asterisk mark indicates accumulation of tubular aggregation (**l**, **n**). **j**, **l** Low; **k**, **m–o**, high magnification. Scale bars, 100 μm (**a**, **i**), 1 μm (**j**, **l**), and 500 nm (**k**, **m–o**). Data are presented as means ± s.e.m. Data shown are representatives of at least three independent experiments. $n = 5$ and 4 for Mib1$^{WT}$ and Mib1$^{\Delta MF}$ mice (**b**); $n = 5$ (**c–g**), $n = 54$, 48, 43, and 68 sections for four mice for Mib1$^{WT}$ mice and 59, 42, 37, and 62 sections for four mice for Mib1$^{\Delta MF}$ mice (**h**), and $n = 4$ and 6 for Mib1$^{WT}$ and Mib1$^{\Delta MF}$ mice (**i–l**), respectively. $\chi^2$ test for trends for (**c**, **g**). Two-tailed Student's $t$ test for (**b**, **d**, **h**). Two-way ANOVA for (**e**, **f**). *$P < 0.05$; **$p < 0.01$; ***$p < 0.001$.

in 16-month-old Mib1$^{\Delta MF}$ muscles indicate that prolonged absence of Mib1 in myofiber results in accelerated age-associated muscle atrophy.

**Impaired muscle function in Mib1$^{\Delta MF}$ mice.** We next evaluated whether a decrease in muscle mass and selective alteration of type 2 glycolytic myofibers lead to impaired muscle function in Mib1$^{\Delta MF}$ mice. To determine the change in muscle function with age, the longitudinal in vivo muscle function analysis was performed. We conducted an in vivo 13-month longitudinal study of skeletal muscle function by measuring whole-limb grip strength

test and endurance test at 3, 9, and 16 months (Fig. 3a). The grip strength of Mib1$^{\Delta MF}$ mice was comparable to those of Mib1$^{WT}$ at an early age. However, with age, the grip strength of 16-month-old Mib1$^{\Delta MF}$ mice significantly declined compared to that of 16-month-old Mib1$^{WT}$ mice (Fig. 3b). To further assess the muscle function, we subjected mice to treadmill running and analyzed running distance and time to exhaustion (Fig. 3c). Sixteen-month-old Mib1$^{\Delta MF}$ mice ran for significantly shorter running distance and time than did Mib1$^{WT}$ mice (Fig. 3d, e). Moreover, regardless of MuSCs and Notch signaling (Supplementary Fig. 1f–h and 2e–g), Mib1$^{\Delta MF}$ mice showed muscle weakness and

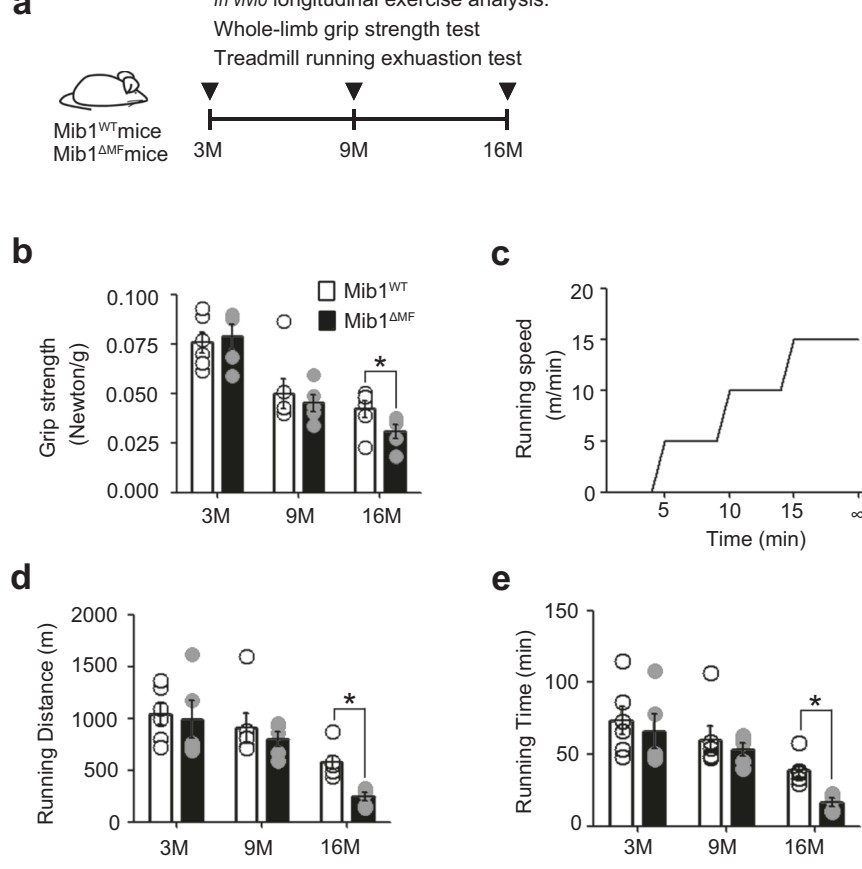

**Fig. 3 Age-associated impaired muscle function in Mib1$^{\Delta MF}$ mice. a** To track changes in skeletal muscle function with age, the longitudinal in vivo muscle function analysis was performed. Mib1$^{WT}$ and Mib1$^{\Delta MF}$ mice were subjected to exercise analysis (whole-limb grip strength test and treadmill running exhaustion test) at 3, 9, and 16 months. **b** Whole-limb grip strength at indicated ages. **c** Scheme of treadmill running. **d, e** Running distance (**d**) and time (**e**) to exhaustion at indicated ages. Data are presented as means ± s.e.m. Data are shown as representatives of at least three independent experiments. $n = 6$ and 5 for Mib1$^{WT}$ and Mib1$^{\Delta MF}$ mice, respectively (**b, d, e**). Two-way ANOVA for (**b, d, e**). *$p < 0.05$.

impaired muscle function with age. Taken together, these results indicate that the loss of Mib1 in myofiber results in the acceleration of age-related muscle atrophy accompanied by rapid loss of muscle function.

**Mib1-dependent regulation of Actn3 via proteasomal degradation pathway.** So far, our results suggest that the selective alteration of type 2 glycolytic myofibers in Mib1$^{\Delta MF}$ mice might be Mib1-mediated intrinsic alterations at the type 2 glycolytic myofibers. To identify the novel Mib1-interacting protein in myofibers, we performed a yeast two-hybrid assay using a complementary DNA (cDNA) library derived from mouse myofibers. Among top three Mib1-binding candidates, we focused on Actn3 and β-enolase (Eno3), which were highly expressed in type 2 glycolytic fibers[15,36] (Supplementary Table1). Immunoprecipitation (IP) experiments showed that Mib1 interacts with Actn3 (Supplementary Fig. 3a). Interaction between Mib1 and Actn3 was further confirmed in GA muscle lysates by endogenous co-IP (Fig. 4a). Actn3 is composed of three domains: actin-binding domain, which binds to actin; spectrin repeats, which forms dimers and interact with various structural and signaling proteins; and EF-hand, which binds to other proteins (Supplementary Fig. 3b)[14]. To identify the binding sites essential for direct interaction between Mib1 and Actn3, we performed the domain mapping assay using various deletion constructs of Actn3 and found that actin-binding domain and spectrin repeat domains of

Actn3 are required for the interaction with Mib1 (Supplementary Fig. 3c).

Next, we investigated the significance and physiological role of Mib1–Actn3 interaction in skeletal muscle. To determine whether Mib1 regulates ubiquitination and proteasome-dependent degradation of Actn3, 293T cells were transfected with Mib1-2× FLAG (Mib1WT), catalytic inactive Mib1 mutant-2× FLAG (Mib1ΔR-ING), hemagglutinin (HA)-ubiquitin, and Actn3-MycHis. After 6 h of incubation with proteasome inhibitor MG-132, cells were subjected to IP using an anti-myc antibody. Mib1WT induced the accumulation of ubiquitination of Actn3, while Mib1ΔRING resulted in diminished ubiquitination of Actn3 (Fig. 4b), suggesting that Mib1 plays a role in the degradation of Actn3. Consistently, the protein expression levels of Actn3 were regulated by Mib1 in a dose-dependent manner (Fig. 4c). In addition, cycloheximide chase (CHX) assay revealed that Mib1 regulates Actn3 via proteasomal degradation pathway (Fig. 4d and Supplementary Fig. 3d). Moreover, Actn3 degraded much faster when Mib1 was co-expressed (Fig. 4d, e), indicating Mib1 is a bona fide E3 ubiquitin ligase for Actn3. Since Actn3 expression levels increased during natural aging process (Fig. 4f, g), we further evaluated whether the ablation of *Mib1* leads to dysregulation and accumulation of Actn3 related to protein degradation and induction of other common E3 ubiquitin ligases (*Atrogin1* and *MuRF*) related to muscle atrophy in vivo. The protein expression levels of Actn3 were significantly increased in 16-month-old Mib1$^{\Delta MF}$ GA muscles compared to Mib1$^{WT}$ GA

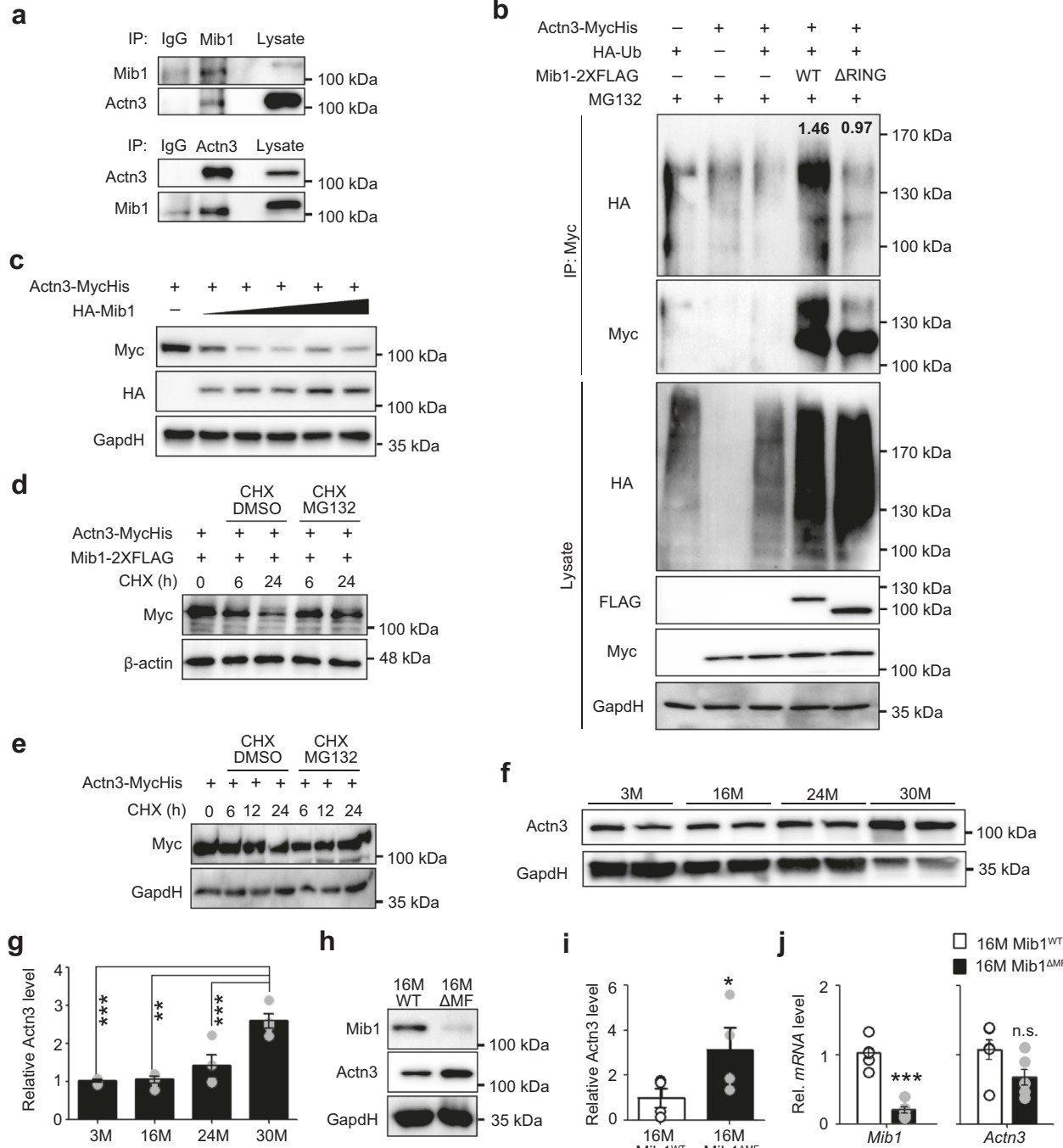

**Fig. 4 Regulation of Actn3 via Mib1-mediated proteasomal degradation pathway in skeletal muscles. a** Endogenous interaction of Mib1 and α-Actn3 (Actn3) in GA muscle lysates. GA muscle lysates from 3-month-old WT mice were subjected to endogenous IP using anti-Mib1 and Actn3 antibody. Mouse and rabbit IgG were used as a nonspecific control. **b** IB analysis of ubiquitination and proteasomal degradation of Actn3 in 293T cells. The 293T cells were transfected with indicated plasmids and treated with 10 μM MG-132 for 6 h. Whole lysates were subjected to IP using anti-Myc, followed by IB analysis. The intensities of HA (Ub) protein of lanes 4 and 5 were 1.46 ± 0.12 and 0.97 ± 0.07, respectively ($p = 0.03$). **c** IB analysis of Mib1-dependent Actn3 protein levels. The 293T cells were transfected with 4 μg of MycHis-tagged Actn3 and 0.5, 2, 4, 8, and 16 μg of HA-tagged Mib1. **d, e** IB analysis of Actn3 protein degradation in 293T cells with (**d**) or without (**e**) Mib1-2× FLAG. 293T cells were transfected with MycHis-tagged Actn3 and/or HA-tagged Mib1 followed by treatment with cycloheximide (CHX) and MG-132 or 0.1% DMSO for the indicated times. **f, g** IB analysis (**f**) and the intensity (**g**) of Actn3 expression in GA muscles of WT mice with indicated ages. **h, i** IB analysis (**h**) and the intensity (**i**) of Mib1 and Actn3 expression in GA muscles of 16-month-old Mib1$^{WT}$ and Mib1$^{ΔMF}$ mice ($p = 0.019$ for **i**). **j** Mib1 and Actn3 mRNA levels in GA muscles ($p = 2.06E{-}05$ for Mib1 and 0.0519 for Actn3). The intensity of protein expression of IB was quantified by densitometry. Data are shown as representatives of at least three independent experiments. Data are presented as means ± s.e.m. $n = 4$ (**f, g**), $n = 6$ (**h–j**) mice per genotypes. One-way ANOVA for (**g**). Two-tailed Student's $t$ test for (**i, j**). *$P < 0.05$; **$p < 0.01$; ***$p < 0.001$; n.s. not significant.

muscles (Fig. 4h–j), indicating posttranslational dysfunction and accumulation of Actn3 in Mib1-deficient skeletal muscles. In addition, there was no change in the expression levels of E3 ubiquitin ligases related to muscle atrophy (Supplementary Fig. 3e), suggesting that muscle atrophy in Mib1$^{\Delta MF}$ mice is solely due to loss of E3 ubiquitin ligase Mib1. These results suggest that Mib1 directly binds to and ubiquitinates Actn3, leading to subsequent proteasome-mediated degradation in skeletal muscle.

**Muscle atrophy of young Mib1$^{\Delta MF}$ mice after chronic moderate exercise.** Since Mib1 regulates sarcomeric protein Actn3 (Fig. 4 and Supplementary Fig. 3c–e), a major component of Z-disk, we questioned whether prolonged dynamic and repetitive contractions of skeletal muscle can induce muscle atrophy even at an early age in the absence of Mib1 in myofiber. We developed a chronic exercise regime of moderate intensity (Supplementary Fig. 4a) that would not cause exercise-induced muscle damage. Prior to chronic exercise, 3-month-old sedentary Mib1$^{WT}$ and Mib1$^{\Delta MF}$ mice (hereafter, Mib1$^{WT-SED}$ and Mib1$^{\Delta MF-SED}$) did not show any apparent abnormality in skeletal muscle and muscle function (Supplementary Fig. 4d–i). Three-month-old mice were then accustomed to chronic exercise (hereafter, Mib1$^{WT-EX}$ and Mib1$^{\Delta MF-EX}$). After 6 weeks of chronic exercise, Mib1$^{\Delta MF-EX}$ mice showed a gradual decrease in hindlimb muscle masses, while Mib1$^{WT-EX}$ mice showed muscle hypertrophy (Supplementary Fig. 4b, j–l). To examine the intrinsic changes of skeletal muscle in Mib1$^{\Delta MF-EX}$ mice, we focused on and analyzed Q muscle, which showed a dramatic reduction in muscle mass after chronic exercise (Fig. 5a–c). Intriguingly, Mib1$^{\Delta MF-EX}$ Q muscle showed a decline in myofiber numbers, particularly type 2 glycolytic fibers (Fig. 5d, e), reflecting muscle atrophic features usually observed in aged mice. Moreover, consistent with increased level of Actn3 in 16-month-old Mib1$^{\Delta MF}$ mice (Fig. 4h, i), Actn3, which was comparable during sedentary state, was also accumulated in Mib1$^{\Delta MF-EX}$ Q muscle after chronic exercise (Supplementary Fig. 4c and Fig. 5f, g), indicating that loss of Mib1 in myofiber leads to accumulation of Actn3 in skeletal muscle and subsequently induce muscle atrophy.

Since chronic exercise can activate muscle MuSCs[37], we analyzed chronically exercised 3-month-old SC$^{WT}$ and SC$^{DTA}$ (hereafter, SC$^{WT-EX}$ and SC$^{DTA-EX}$) mice to examine whether exercise-induced muscle atrophy is caused by the depletion of MuSCs. SC$^{DTA-EX}$ mice showed comparable muscle mass and even increased Q muscle mass after chronic exercise (Supplementary Fig. 5a). Normal muscle morphology of SC$^{DTA-EX}$ mice (Supplementary Fig. 5b–e) indicates that the depletion of muscle stem cells is not responsible for chronic exercise-induced muscle atrophy observed in Mib1$^{\Delta MF-EX}$ mice. Taken together, these results suggest that chronic exercise or dynamic use of skeletal muscle can hasten or aggravate muscle atrophy in young Mib1$^{\Delta MF-EX}$ mice.

**Amelioration of chronic moderate exercise-induced muscle atrophy in young Mib1$^{\Delta MF}$ mice by downregulation of Actn3.** Since Actn3 was accumulated in Q muscles of 3-month-old Mib1$^{\Delta MF}$ mice after chronic exercise (Fig. 5f, g), we expected that the prevention of Actn3 accumulation in skeletal muscle prior to chronic exercise could ameliorate the chronic exercise-induced muscle atrophy. To address the effect of the prevention of Actn3 accumulation in 3-month-old Mib1$^{\Delta MF-EX}$ mice, lentiviral short hairpin RNA (shRNA) targeting control shRNA (shCtrl) or Actn3 (shActn) was intramuscularly (i.m.) administered to both sides of Q muscles (Fig. 5h). One week after i.m. injection of shCtrl or shActn into 3-month-old Mib1$^{\Delta MF}$ mice, mice were subjected to chronic exercise for 6 weeks, and the downregulation

of Actn3 level was confirmed by IHC staining and western blotting against Actn3 (Supplementary Fig. 6a, b). The Q muscle mass of shActn3-injected and exercised Mib1$^{\Delta MF}$ mice (hereafter, $\Delta$MF-EX + shActn3) was significantly increased compared to those of Mib1$^{\Delta MF-EX}$ and shCtrl-injected and exercised Mib1$^{\Delta MF}$ mice (hereafter, $\Delta$MF-EX + shCtrl) (Fig. 5i). Further analysis of Q muscles showed that the number of myofibers, particularly type 2 glycolytic myofibers, were significantly increased in $\Delta$MF-EX + shActn3 compared to those of Mib1$^{\Delta MF-EX}$ and $\Delta$MF-EX + shCtrl mice (Fig. 5d, e, j, k). $\Delta$MF-EX + shActn3 mice showed a larger mean CSA of type 2 glycolytic myofibers compared to those of $\Delta$MF-EX + shCtrl mice (Supplementary Fig. 6c). The Q muscles of $\Delta$MF-EX + shActn3 showed the high and low intensity of Actn3 in outer and inner part, respectively, due to uneven delivery of lentiviral shRNA within skeletal muscles (Fig. 5l). In that, the corresponding CSAs for $\Delta$MF-EX + shCtrl and $\Delta$MF-EX + shActn3 (shCtrl$^{inner}$ vs. shActn3$^{inner}$, and shCtrl$^{outer}$ vs. shActn3$^{outer}$) (Fig. 5l, m) were further analyzed. Consistent with mean CSA of whole type 2 glycolytic myofibers (Supplementary Fig. 6c), the CSAs of inner part of $\Delta$MF-EX + shActn3 (shActn3$^{inner}$) myofibers were significantly larger than the that of corresponding part of $\Delta$MF-EX + shCtrl (shCtrl$^{inner}$) [1470.97 ± 14.01 μm$^2$ (shCtrl$^{inner}$) vs. 1911.01 ± 16.03 μm$^2$ (shActn3$^{inner}$)], while similar in that of outer part [1969.83 ± 16.03 μm$^2$ (shCtrl$^{outer}$) vs. 1984.29 ± 21.17 μm$^2$ (shActn3$^{outer}$)] (Fig. 5n), indicating that the downregulation of Actn3 can prevent the loss of type 2 glycolytic myofibers and mitigate the chronic exercise-induced muscle atrophy. Collectively, these data highlight that the regulation of Actn3 levels to an optimal ratio in skeletal muscle might be a potential therapeutic strategy to alleviate or prevent muscle atrophy.

**Disturbed Mib1–Actn3 axis in human skeletal muscle with age.** Given that the Mib1–Actn3 axis plays an important role in age-associated muscle atrophy, we further evaluated the clinical relevance of the Mib1–Actn3 axis in human skeletal muscles with age. We analyzed the vastus lateralis muscles of middle-aged (<60 years) and aged groups (>60 years). The aged group was further divided into aged and sarcopenic groups according to Asian Working Group Society (AWGS) criteria (Supplementary Table 2 and Supplementary Fig. 7a)[38]. Consistent with Mib1 expression in mouse skeletal muscle (Fig. 1a, b), the expression level of Mib1 also significantly decreased in human skeletal muscles of the aged and sarcopenic groups compared to the middle-aged group (Fig. 6a, b). As Mib1 decreases, Actn3 protein level was increased in the aged group (Fig. 6a, c). Interestingly, Actn3 protein level was significantly decreased in the sarcopenic group, in which Mib1 expression was further decreased compared to that of aged group (Fig. 6a–c). The decreased level of Actn3 might be due to either Actn3 polymorphism or the cumulative loss of myofibers, which occurred preferentially in type 2 glycolytic myofibers, in age-associated muscle atrophy[39]. About 20% of the global population have Actn3 R577X polymorphism[15]. To examine whether Actn3 R577X polymorphism is implicated in decreased level of Actn3 in sarcopenic group, we conducted single-nucleotide polymorphism genotyping and found that there were no Actn3 polymorphism in all groups (Supplementary Fig. 7b). Since Actn3 is predominantly expressed in type 2 glycolytic myofiber[14] and the loss of type 2 glycolytic myofiber is a hallmark of sarcopenia[1], we examined the number of type 2 glycolytic myofibers in each group. Expectedly, the numbers of type 2 glycolytic myofibers were significantly decreased in sarcopenic group (Fig. 6d, e), further indicating that the decreased level of Actn3 in sarcopenic group might be due to the accumulation of Actn3 and subsequent loss of type 2 glycolytic myofibers. Collectively, our finding suggests the importance of the Mib1–Actn3 axis

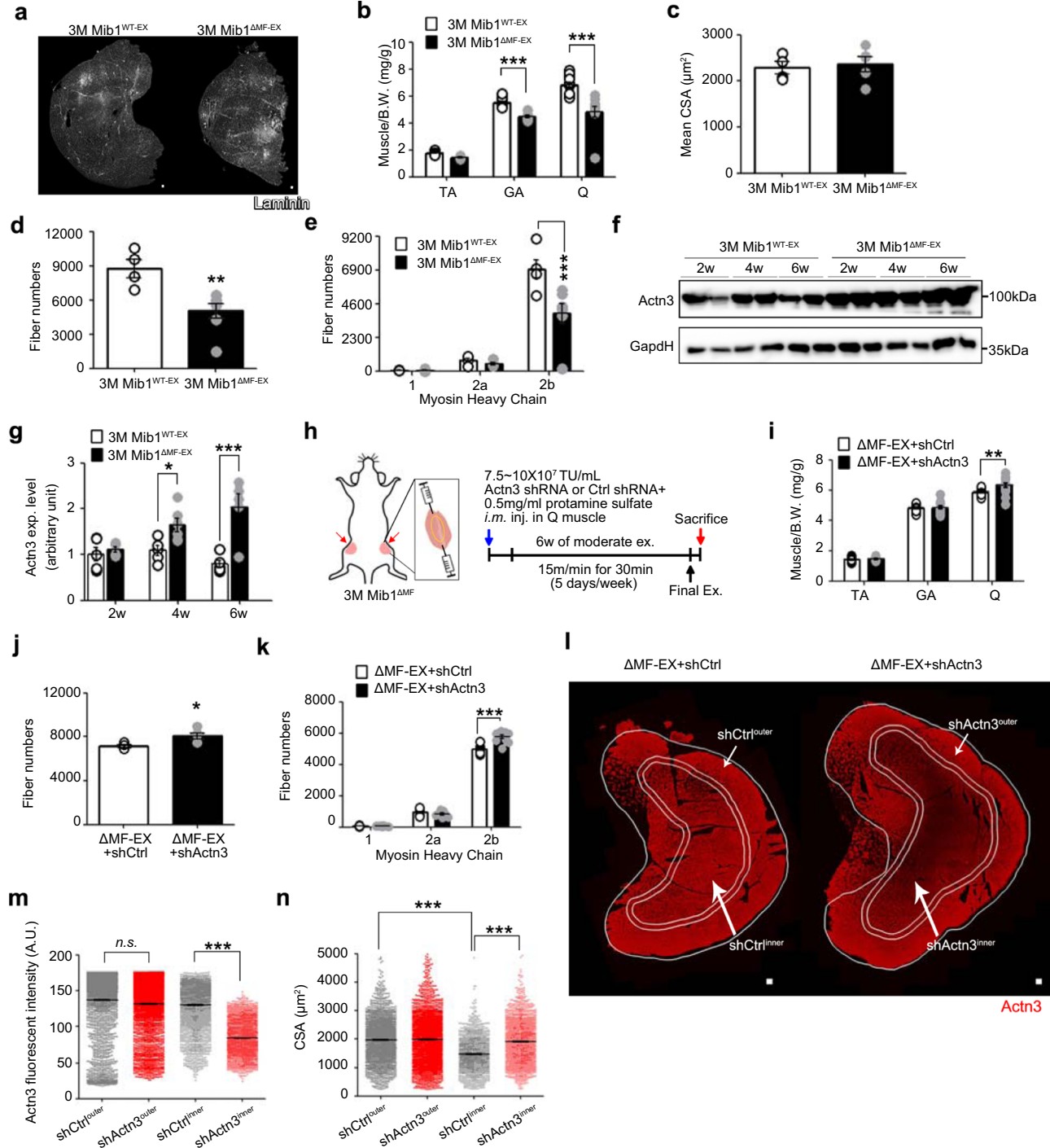

in the skeletal muscle maintenance with age and unravels one of mechanisms implicated in the cause of sarcopenia.

## Discussion

This study uncovered the crosstalk between Mib1 and skeletal muscle maintenance with age, which paves the way toward the understanding of how the loss of Mib1 concomitantly contributes to age-associated muscle atrophy, and subsequent loss and/or atrophy of type 2 glycolytic myofibers. We unravel two important properties of Mib1 in the maintenance of skeletal muscle. First, age-associated decrease in Mib1 levels consequently promotes age-associated muscle atrophy. Second, Mib1 plays a significant role in the maintenance of skeletal muscle, particularly type 2

glycolytic myofibers, by directly binding to and ubiquitinating Actn3, which is predominantly expressed in type 2 glycolytic myofibers. Loss of Mib1 in myofiber results in the accumulation of Actn3 and subsequent alteration of type 2 glycolytic myofibers, thereby accelerating age-associated muscle atrophy. Furthermore, the amelioration of muscle atrophy by downregulation of Actn3 represents Actn3 as a promising therapeutic target of age-associated muscle atrophy. Collectively, we concluded that the Mib1–Actn3 axis is crucial in skeletal muscle maintenance with age and our findings will provide fundamental insights into how the age-associated decline in Mib1 levels leads to age-associated muscle atrophy and alteration of type 2 glycolytic myofibers (Fig. 6f).

**Fig. 5 Induction or amelioration of chronic exercise-induced muscle atrophy of young Mib1$^{\Delta MF}$ mice. a** Representative images of IHC staining for laminin (white). **b** Relative hindlimb muscles to body weights. **c** Mean CSA of Q muscles. **d, e** Quantification of whole myofiber numbers (**d**) and myofiber numbers by fiber types (**e**) ($p = 0.0063$ for **d**). **f, g** IB analysis of Actn3 from insoluble muscle lysates from GA muscles of exercised Mib1$^{WT}$ (Mib1$^{WT-EX}$) and exercised Mib1$^{\Delta MF}$ (Mib1$^{\Delta MF-EX}$) mice (**f**) and intensity of Actn3 (**g**) at indicated times. **h** Three-month-old Mib1$^{\Delta MF}$ were i.m. injected with lentiviral shRNAs targeting control shRNA (shCtrl, hereafter $\Delta$MF-EX + shCtrl) or Actn3 (shActn3, hereafter $\Delta$MF-EX + shActn3) to both sides of Q muscles. The local injection of shRNA into Q muscle is shown in yellow dotted line. A week later, mice were subjected to chronic exercise. **i** Relative hindlimb muscles to body weights. **j, k** Quantification of whole myofiber numbers (**j**) and myofiber numbers by fiber types (**k**) ($p = 0.0039$ for **j**). **l** Representative images of IHC staining for Actn3. The IHC images were taken simultaneously with the same light setting, exposure time, and magnification. Actn3 intensity within shActn3-injected Q muscle was divided into high (outer part; area between single and double line) and low (inner part; surrounded by double line). Corresponding outer and inner parts were shown in shCtrl-injected Q muscles. **m** Actn3 fluorescent intensity of outer and inner part of $\Delta$MF-EX + shCtrl and $\Delta$MF-EX + shActn3. Note that Actn3 intensity of inner part of $\Delta$MF-EX + shActn3 ($\Delta$MF-EX + shActn3$^{inner}$) is significantly lower than corresponding inner part of $\Delta$MF-EX + shCtrl ($\Delta$MF-EX + shCtrl$^{inner}$). **n** CSA of outer and inner part of $\Delta$MF-EX + shCtrl and $\Delta$MF-EX + shActn3. Note that mean CSA of shActn3$^{inner}$ is larger than corresponding part of shCtrl$^{inner}$. Please note that CSA of outer part of skeletal muscle is generally larger than inner part. Scale bars, 100 μm (**a**) and 125 μm (**l**). The intensity of protein expression of IB was quantified by densitometry. Data are presented as means ± s.e.m. Data are shown as representatives of at least three independent experiments. $n = 6$ and 5 for Mib1$^{WT}$ and Mib1$^{\Delta MF}$ mice, respectively (**b**). $n = 4$ and 5 for Mib1$^{WT}$ and Mib1$^{\Delta MF}$ mice, respectively (**c, d, e, g, j**). $n = 8$ and 7 for Mib1$^{WT}$ and Mib1$^{\Delta MF}$ mice, respectively (**i, k**). $n = 3169$, 3749, 2662, and 2827 myofibers for shCtrl$^{outer}$, shActn3$^{outer}$, shCtrl$^{inner}$, and shActn3$^{inner}$, respectively (**m**). $n = 2218$, 2624, 1597, and 1696 myofibers for shCtrl$^{outer}$, shActn3$^{outer}$, shCtrl$^{inner}$, and shActn3$^{inner}$, respectively (**n**). One-way ANOVA for (**m, n**). Two-way ANOVA for (**b, e, i, k**). Two-tailed Student's $t$ test for (**c, d, j**). *$P < 0.05$; **$p < 0.01$; ***$p < 0.001$.

According to Garton et al.[18], the conservation of the optimal level of Actn3 in type 2 glycolytic myofibers seems critical in maintaining skeletal muscle. The overexpression of Actn3 in tibialis anterior muscle results in damaged muscle structure and abnormal muscle function. Interestingly, muscle damage seems to correlate with the level and the duration of excessive expression of Actn3. When a high dose of Actn3 was i.m. injected, there was no sign of muscle damage until 2 weeks, indicating that excessive expression of Actn3 above a certain threshold in whole or certain regions of muscle can have toxic effects on skeletal muscle. Although Garton et al.[18] provide detrimental effects of accumulation of Actn3 in skeletal muscles, the mechanism by which high levels of Actn3 leads to muscle damage and dysfunction has been still elusive. Indeed, Z-disk is composed of two to six layers of Z-disks[40]. These variable layers of Z-disk might allow the sarcomeric structure to bear gradual accumulation of Actn3 up to a certain threshold. In this study, Actn3 was accumulated in 16-month-old Mib1$^{\Delta MF}$ skeletal muscle (Fig. 4h, i). This excess and progressive accumulation of Actn3 might overburden the variable layers of Z-disks, further developing into disorganized Z-disk, accumulation of tubular aggregations and aberrant membrane structure in type 2 glycolytic myofibers. As further accumulation occurs, the selective alteration of type 2 glycolytic myofibers may eventuate, leading to the acceleration of age-associated muscle atrophy. Although it remains elusive whether dose-dependent regulation of Actn3 in myofibers can ameliorate the selective alteration of type 2 glycolytic myofibers in normal aging, we suggest a mechanism by which type 2 glycolytic myofibers are maintained throughout the lifetime via Mib1–Actn3 axis.

Skeletal muscle exhibits phenotypic plasticity and undergoes continuous remodeling throughout life in response to various environmental and physiological cues[41]. In that, continuous skeletal muscle remodeling is instrumental to maintain skeletal muscle homeostasis and myofiber functional integrity. However, with age, tissue and protein homeostasis is perturbed leading to alteration of proteostasis, accumulation of damaged proteins and age-related loss of skeletal muscle mass and function[3,4]. Recent studies showed that accumulation of protein aggregates and/or sarcomeric protein develop into age-associated muscle degeneration, myopathology, and contractile dysfunction[3,42,43]. For example, the overexpression of myotillin, a component of Z-disk, or desmin, intermediate filaments which integrate Z-disk, in skeletal muscle leads to the accumulation of myofibrillar and tubular aggregates with age, abnormal sarcolemmal integrity,

muscle atrophy, and subsequent muscle degeneration and impaired contractile function[44]. These results suggest that the dysregulation and toxic accumulation of sarcomeric and/or myofibrillar proteins can lead to the pathological condition of skeletal muscle that contributes to muscle atrophy and abnormal function. In case of Mib1$^{\Delta MF}$ mice, the accumulation of Actn3 might result in the disturbed assembly of sarcomeric proteins such as titin, desmin, vinculin, and Z-band alternatively spliced PDZ motif-containing protein that are known to bind with Actn3[17]. Among these proteins, titin and desmin are one of the major Z-disk proteins involved in sarcomere assembly, mechanosensor, and force transducer function[45]. The disorganized Z-disk in Mib1ΔMF mice suggests the possibility of impaired muscle contractile activity. For example, mutated desmin adversely affects the endurance function of mice because of low force generation and transmission[46]. In addition, when titin is impaired, the force generation and transmission are also impaired since titin is an elastic element that contributes to passive tension on a force–length relationship in myofibers[47]. In that, the maintenance of sarcomeric integrity and assembly, especially Z-disk, is important to bear contraction-induced mechanical stress[48]. Considering the interaction between Actn3 and other sarcomeric proteins in Z-disks, the accumulation of Actn3 might lead to abnormal crosslinking or assembly of sarcomeric proteins, which are essential for force generation and transmission. Thus, the coordinated regulation of myofibrillar and sarcomeric protein is essential to maintain protein quality throughout a lifetime. Although several studies have investigated the effects and formation of abnormal protein aggregates in skeletal muscle at an advanced age, the mechanisms responsible for the formation of protein aggregates or accumulation of sarcomeric proteins with age are still largely unknown. Thus, our findings on Mib1-dependent accumulation of Actn3 with age will expand the understanding of how age-associated regulation of sarcomeric proteins is implicated in age-associated muscle atrophy.

During aging, the apoptotic cell death pathway by various types of age-related cellular stresses such as proteotoxic stress, oxidative stress, or protein misfolding can contribute to the loss of myofibers[4,49,50]. Although several mechanisms involved in apoptosis in myofibers are reported, it has been still elusive how selective loss of type 2 glycolytic myofibers occur with age. In this study, we showed that Actn3, predominantly expressed in type 2 glycolytic myofibers[14], was accumulated in total muscle lysates and an insoluble fraction of muscle lysates (Fig. 4h, f), indicating

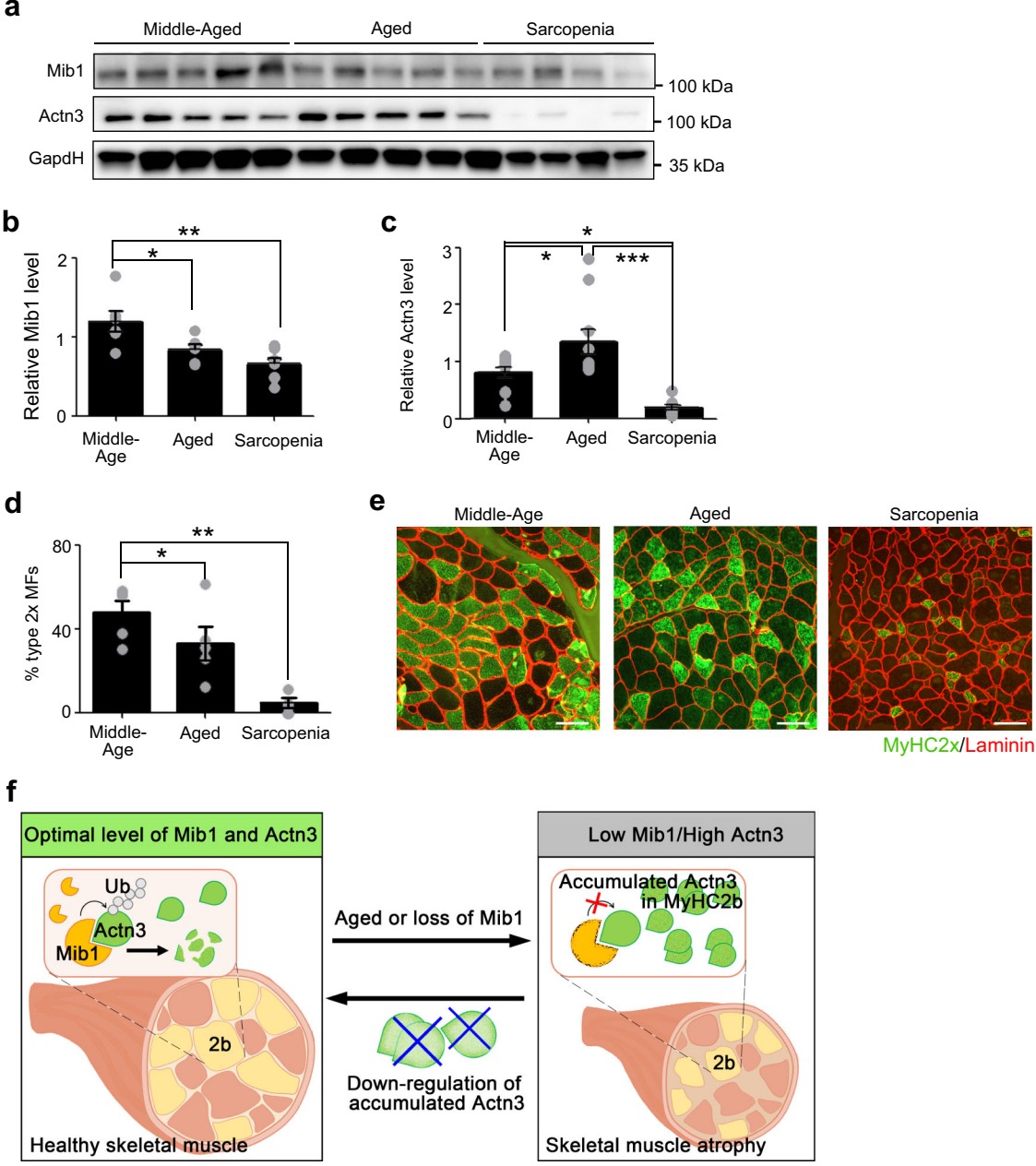

**Fig. 6 Disturbed Mib1–Actn3 axis in human skeletal muscle with age. a–c** IB analysis (**a**) of Mib1 and Actn3 in vastus lateralis muscles of middle-aged (<60 years), aged (>60 years), and sarcopenia (>60 years and meet AGWS criteria), and quantification of Mib1 (**b**) and Actn3 (**c**). Please see Supplementary Fig. 7a for flow chart of selection of human subjects. The intensity of Mib1 and Actn3 expression at indicated group was quantified by densitometry. **d**, **e** Quantification of the percentage of type 2× glycolytic myofibers (**d**) and **e** representative images of IHC staining for MyHC2x (green; type 2× glycolytic myofibers) and laminin (red). Scale bars, 100 μm (**e**). Data are presented as means ± s.e.m. Data are shown as representatives of at least three independent experiments. $n = 6$ for middle-aged and aged group, and 8 for sarcopenic group (**a–c**). $n = 5$ for the middle-aged and aged group, and 4 for sarcopenic group (**d**). One-way ANOVA for (**b–d**). *$P < 0.05$; **$p < 0.01$; ***$p < 0.001$. **f** A proposed model. In healthy myofibers, Mib1 regulates Actn3, a Z-disk protein highly expressed in type 2 glycolytic myofibers, in a proteasome-dependent manner to maintain the optimal levels of Actn3. However, the age-associated changes in or loss of Mib1 in myofibers lead to the accumulation of Actn3 accompanied by the alteration of type 2 glycolytic myofibers, muscle atrophy, impaired muscle function, and, consequently, an acceleration of age-associated muscle atrophy. When the excessive accumulation of Actn3 is alleviated by the downregulation of Actn3 in myofibers, the muscle atrophy is ameliorated, suggesting that Actn3 can be a promising therapeutic target of age-associated muscle atrophy.

that increase in Actn3 protein can lead to formation of large insoluble aggregates in skeletal muscle. Indeed, Actn3 forms antiparallel homodimers that cross-link or interact with sarcomeric proteins in the Z-disks[51]. The dimerization has a high propensity to form protein aggregation, resulting in toxic aggregates that can cause cytotoxicity or proteotoxicity[52]. Interestingly,

according to the prediction algorithm Waltz, which predicts the amyloid regions[53], there are two highly amyloidogenic regions, from amino acids 139 to 153 and from amino acids 688 to 699, in Actn3, indicating that Actn3 can form either amorphous aggregates or amyloid fibrils. Thus, the increased dimeric Actn3 can act as seeds for further aggregation. Although it is still elusive how

dimeric form of Actn3 acts as seeds for aggregates, as reported, the accumulation of protein aggregates such as dimeric Actn3 aggregates in skeletal muscle can lead to toxic accumulation and subsequently proteotoxic stress, which can activate apoptotic pathway[4,54]. Taken together, our finding on accumulation of Actn3 aggregates in skeletal muscle suggests the possible mechanism related to loss of selective type 2 glycolytic myofibers via apoptotic pathway. Among the proteolytic pathways, autophagy is also involved in the degradation of proteins, organelles, and protein aggregates[4]. The sustained accumulation of protein aggregates in Mib1$^{\Delta MF}$ mice can overburden autophagy–lysosomal activity with increased autophagosome synthesis accompanying excessive accumulation or production of the autophagosome. Eventually, it can result in compromised autophagy–lysosomal activity and further contribute to proteotoxictiy[55,56]. Since Actn3 has the potential to form aggregates and/or amyloid structure, the sustained accumulation of Actn3 aggregates can induce excessive autophagy and overburden the autophagic capacity in Mib1$^{\Delta MF}$ mice. Further studies are required to examine whether the Actn3 forms amorphous aggregates or amyloid fibrils, which compromise the function of the autophagy–lysosomal degradation pathway in Mib1$^{\Delta MF}$ mice.

As aforementioned, to maintain skeletal muscle maintenance, it is fundamental to have a balance between protein degradation and protein turnover. Among E3 ubiquitin ligases which play a critical role in the maintenance of protein homeostasis, MuRF, atrogin-1/MafBx and Trim32 play a significant role in muscle atrophy[57–60]. Although these E3 ubiquitin ligases are implicated in the degradation of myofibrillar and sarcomeric proteins, the loss of function studies showed the prevention of muscle atrophy under muscle atrophic conditions such as denervation, unloading, fasting or diabetes[57,58,61]. However, recent studies showed that skeletal muscle deficient of E3 ubiquitin ligases can develop skeletal muscle myopathy due to the accumulation of sarcomere proteins. The loss of MuRF1, MuRF2 and/or MuRF3, results in myopathy due to abnormal accumulation of myofibrils in myofibers[60,62]. In addition, skeletal muscle-specific deletion of Cullin-3 E3 ligases (Cul3) results in musculature defects, alteration of sarcomeric protein expressions and disorganized sarcomere[63], indicating E3 ubiquitin ligase is also responsible for UPS-dependent sarcomeric protein turnover to maintain skeletal muscle maintenance. However, it is still elusive which E3 ubiquitin ligase is implicated in skeletal muscle maintenance. In 16-month-old Mib1$^{\Delta MF}$ mice, there were no significant changes in the expression levels of muscle-specific E3 ubiquitin ligases (*Atrogin-1* and *MuRF1*) (Supplementary Fig. 3e). Although MuRF1 and Atrogin-1 are well-known E3 ubiquitin ligases implicated in skeletal muscle atrophy, at least ten or more uncharacterized E3 ubiquitin ligases play roles in skeletal muscle[20]. Moreover, Lodka et al. showed increased ubiquitin–proteasome proteolytic activation in *MuRF2* and *MuRF3* double knockout skeletal muscle[62], suggesting that the ubiquitin–proteasome proteolytic pathway can be activated by other E3 ubiquitin ligases in the absence of MuRF2 and MuRF3. In that, unknown E3 ubiquitin ligase-dependent ubiquitin–proteasome proteolytic pathway beside Atrogin-1 and MuRF1 may be implicated in muscle atrophy of Mib1$^{\Delta MF}$ mice. Taken together, our work identifying Mib1 as a novel E3 ubiquitin ligase for Actn3 provides a mechanism for phenotypic analysis on Mib1$^{\Delta MF}$ mice which showed an acceleration of muscle wasting due to accumulation of Actn3 and abnormal sarcomeric structure. Furthermore, we showed that optimal regulation of Actn3 by Mib1-mediated proteasomal degradation is essential for skeletal muscle maintenance.

Among several physiological factors contributing to age-associated muscle atrophy, sex steroid hormones are one of the major factors known to change with age. Considering the anabolic effects of sex hormones on skeletal muscle, many efforts have been made to understand the effects of sex hormones on age-associated muscle atrophy[64,65]. Previously, we showed that Mib1 is induced in myofibers by sex hormone at puberty and is decreased in 3-month-old gonadogrophin-releasing hormone-deficient hypogonadal (*Gnrh1$^{hpg/hpg}$*) mice[66], in which sex hormones are reduced[29]. As both Mib1 and sex hormones decrease with age, we speculated that not only decreases in sex hormones with age, but also the prolonged reduction of sex hormones might result in a decrease in Mib1 expression and acceleration of muscle atrophy. Indeed, we observed a decline of Mib1 expression in GA muscle with age (Fig. 1a, b), suggesting that the sex hormone–Mib1 axis might be implicated in age-associated muscle atrophy. In this study, we showed that age-dependent expression of Mib1, which is also regulated by sex hormones[29], in skeletal muscle is essential to maintain type 2 glycolytic myofibers and skeletal muscle with age, thereby suggesting that the sex hormone–Mib1 axis is possibly implicated in skeletal muscle maintenance. However, the mechanism by which sex hormones is implicated in the maintenance of skeletal muscle and type 2 glycolytic myofibers still remained unknown and to be determined. In that, further studies on the implications of sex hormones and Mib1 in sarcopenia using pharmacologically or genetically modified mice are required to reveal the putative correlation between sex hormones and Mib1 with age.

Since muscle stem cells are indispensable for muscle regeneration[67], it has long been believed that loss of muscle stem cells with age might cause sarcopenia. However, unlike general belief, Fry et al.[31] reported that the reduction of muscle stem cells does not accelerate or exacerbate sarcopenia. Aged muscle stem cell-depleted mice showed comparable muscle size, fiber type composition, and muscle function compared to wild-type aged mice[31]. In accordance with the previous report, our study also showed that loss of muscle stem cells do not contribute to sarcopenia (Supplementary Fig. 1). Not only muscle stem cells but Notch signaling have gotten much attention in muscle atrophy. Several studies report that Notch signaling diminished with age and restoration of Notch signaling via heterochronic parabioses or forced activation of Notch signaling in aged skeletal muscles improve impaired regenerative capacity of aged skeletal muscles[68,69]. Despite the correlation between decreases in Notch signaling and sarcopenia, no studies to date have examined whether sarcopenia is accelerated or exacerbated in Notch-deficient mice. As muscle stem cells are Notch receiving cells, we used muscle stem cell-specific Notch-deficient (Notch$^{\Delta SC}$) mice to determine the relationship between Notch signaling and sarcopenia. Consistent with previous studies, Notch$^{\Delta SC}$ mice showed loss of muscle stem cells (Supplementary Fig. 2b). However, intriguingly, 16-month-old Notch$^{\Delta SC}$ mice showed no sign of muscle atrophy and even greater muscle functional capacity compared to Mib1$^{\Delta MF}$ mice (Supplementary Fig. 2e–g), indicating that Notch signaling is not associated with sarcopenia. Thus, our study provides convincing evidence that loss of Notch signaling in muscle stem cells does not contribute to sarcopenia. Consequently, we, for the first time as far as we know, showed that Mib1 plays Notch-independent role in the maintenance of skeletal muscle with age.

Our findings suggest that the Mib1–Actn3 axis is critical to maintaining skeletal muscle throughout the lifetime. The decrease in Mib1 or accumulation of Actn3 in skeletal muscle with age represents as a therapeutic target to treat or ameliorate age-associated muscle atrophy. Due to difficult to study the natural and gradual aging in mouse, several mouse models with accelerated or premature aging phenotypes have been generated[70]. Although age-associated muscle atrophy mouse models showed

age-related loss of muscle mass and function, these genetically manipulated mouse models to mimic age-associated muscle atrophy at an early age do not solely demonstrate the age-associated muscle atrophic characteristics observed in skeletal muscles. Some mouse models showed other pathological features, including kidney dysfunction, osteoporosis, degenerative joint disease, and tumors, which also can have adverse effects on skeletal muscle[70]. Although some aging mouse models showed similar aging phenotypes of Mib1$^{\Delta MF}$ mice such as tubular aggregates[71], muscle atrophy, and impaired muscle function[72], the causality of age-associated muscle atrophy is different. In the case of Mib1$^{\Delta MF}$ mice, the age-associated muscle atrophy is originated from the loss of Mib1 in myofibers unlike systemic aging of aging mouse model. In that, the development of a valid mouse model for age-associated muscle atrophy is necessary to examine the effects of aging solely on skeletal muscle. In this study, we showed a gradual loss of muscle mass and function with age using myofiber-specific Mib1-floxed mouse models. Our mouse model did not show any signs or symptoms of a complication, indicating that Mib1$^{\Delta MF}$ mice can be used as an age-associated muscle atrophy model to study the intrinsic change of skeletal muscle with age. Moreover, chronic exercise-induced muscle wasting in young Mib1$^{\Delta MF}$ mice reveals the significance and necessity of adjustable exercise strategies for the elderly. As expectations on healthy aging rise, our results will broaden our understanding of therapeutic strategy for age-associated muscle atrophy. Collectively, this study will shed a light on the mechanism related to the Mib1–Actn3 axis in skeletal muscle maintenance throughout a lifetime.

## Methods

**Animals**. All male mice from C57BL/6J background were housed in a 12-h light/12-h dark cycle at room temperature (RT) 22 °C with 40–60% humidity, and fed ad libitum. *Mib1$^{f/f}$* mice were previously generated[22]. *MCK-Cre*, *Pax7-Cre$^{ER}$*, *Rosa-DTA* transgenic mice and *Notch1$^{f/f}$*, *Notch2$^{f/f}$*, and *Rbpjκ$^{f/f}$* mice were purchased from the Jackson Laboratory (Bar Harbor, ME, USA). *MCK-Cre* and *Pax7-Cre$^{ER}$* mouse lines were used for myofiber or muscle stem cell-specific deletion studies, respectively. Tamoxifen (20 mg mL$^{-1}$ in corn oil) (Sigma-Aldrich, St. Louis, MO, USA) was orally administered daily for 5 consecutive days (160 mg kg$^{-1}$ body weight).

**Human participants and skeletal muscle tissue sample collection**. Incised and marginally resected vastus lateralis muscle (average 0.5 ± 0.1 g) tissue samples were obtained from female patients who underwent hip joint surgery. Subjects were divided into three groups: middle-age (<60 years), aged (>60 years), and advanced-age groups (>60 years and meet Asian Working Group Society criteria). The flow chart showing the selection of study subjects was presented in Supplementary Fig. 7a. Subjects' characteristics were summarized in Supplementary Table 2. Whole-body dual-energy X-ray absorptiometry, grip strength test, and SARC-F questionnaire were performed before the surgery. SARC-F is a simple screening tool to rapidly diagnose sarcopenia[73,74] and is composed of five questions: (Q1) strength, (Q2) assistance in walking, (Q3) standing from a chair, (Q4) climbing stairs, and (Q5) falling down.

**Study approval**. All experimental protocols involving animal studies and human subjects received ethical approval. All animal procedures conformed to the Guide for the Care of Use of Laboratory Animals and were conducted with the approval of the Institutional Animal Care and Use Committee at Seoul National University, Korea. All procedures associated with the human research was approved by the institutional review board (Chung-Ang University Hospital Institutional Review Board) of Chung-Ang University College of Medicine (IRB number: 1981-007-383) and performed in accordance with relevant guidelines and regulations. Subjects provided written informed consent prior to participation, and written informed consent was obtained from all participants.

**Plasmids, shRNA, and cell transfections**. The full-length and deleted constructs of mouse Actn3 were subcloned into the pcDNA3.1-mycHis vector. N-terminal HA- or C-terminal 2× FLAG-tagged full-length and deleted construct of mouse Mib1 were subcloned into the pcDNA3 or pcDNA3.1 vector. HA-tagged Ub was subcloned into pcDNA3 vector. Transfection of plasmids was conducted using metafectene (Biontex Laboratories GmbH, Munich, Germany) or poly-ethyleneimine (Polysciences, Warrington, PA, USA) following the manufacturer's

instructions. The HEK293T cell line was obtained from ATCC and was cultured at 37 °C in Dulbecco's modified Eagle's medium with 10% fetal bovine serum and antibiotics. The cell was regularly tested for mycoplasma contamination.

**Lentivirus packaging and in vivo delivery**. Lentiviral shRNA plasmids for targeting Actn3 were purchased from Dharmacon and control shRNA was generated[75] (Supplementary Table 3). The shRNA (5 μg), psPAX2 (3.75 μg), and pMD2.G (1.25 μg) plasmids were cotransfected into HEK293T cells using poly-ethyleneimine. The culture supernatants containing virus particles were harvested 72 h post transfection, filtered through 0.45 μm filters and concentrated using a Lenti-X concentrator (Takara, Kyoto, Japan). The lentiviral particles with a final titer of 7.5–10 × 10$^7$ TU mL$^{-1}$ were i.m. injected into Q muscles in the presence of 0.5 mg mL$^{-1}$ protamine sulfate.

**Single myofiber isolation**. For single myofiber isolation, dissected hindlimb muscles were enzymatically digested with collagenase type I (2 mg mL$^{-1}$, Gibco Invitrogen, Grand Island, NY) for 60 min at 37 °C. Digested muscles were blocked in Dulbecco's modified Eagle's medium and 10% horse serum (Hyclone, Logan, UT, USA). The single myofibers were released by gentle trituration. Undamaged and noncontracted single myofibers were then washed with phosphate-buffered saline (PBS) several times and collected for yeast two-hybrid screening assay.

**TEM analysis**. Q muscles were fixed with 3% glutaraldehyde in 0.1 M cacodylate buffer (pH 7.2) containing 0.1% CaCl$_2$ for 3 h at RT. Fixed muscle tissues were washed five times with 0.1 M cacodylate buffer at 4 °C and then postfixed with 1% OsO$_4$ in 0.1 M cacodylate buffer containing 0.1% CaCl$_2$ for 2 h at 4 °C. After rinsing with cold distilled water, they were dehydrated slowly with an ethanol series and propylene oxide at 4 °C. The samples were embedded in Embed-812 (EMS, PA, USA). After polymerization of the resin at 60 °C for 36 h, serial sections were cut with a diamond knife on an ULTRACUT UC7 ultramicrotome (Leica, Wetzlar, Germany) and mounted on formvar-coated slot grids. Sections were stained with 4% uranyl acetate for 10 min and lead citrate for 7 min. Sections were observed using a Tecnai G2 Spirit Twin TEM (FEI Company, Hillsboro, OR, USA).

**Actn3 R577X genotyping**. Genomic DNA was isolated from human skeletal muscles. Actn3 R577X genotyping was performed using the following primers[76]: ACTN3-F 5′-CTGTTGCCTGTGGTAAGTGGG-3′ and ACTN3-R 5′-TGGTCA-CAGTATGCAGGAGGG-3′. The amplicons were then enzymatically digested with *Dde*I as R577X allele has a restriction site: fragments of 205 and 86 bp in the digestion of the R577 allele, while fragments of 108, 97, and 86 bp in the digestion of the 577X allele. The size separation of DNA fragments was performed by agarose gel electrophoresis using 2.5% agarose gel.

**RNA isolation and quantitative real-time polymerase chain (qRT-PCR)**. Total RNA was extracted from freshly isolated GA muscles using TRI Reagent (Invitrogen) and analyzed by qRT-PCR using Rotor-Gene-Q pure-detection software. The first-strand complementary DNA was performed synthesized from 1 μg of RNA using Promega's RT system according to the manufacturer's instructions (Promega, Madison, WI, USA). qRT-PCR (Qiagen) was performed with SYBR Green technology (SYBR Premix Ex Taq, Qiagen) using specific primers against indicated genes. Expression levels were determined using the 2$^{-\Delta\Delta Ct}$ method and normalized to the housekeeping gene *GAPDH*. Primers are listed in Supplementary Table 3.

**Immunoblot and co-IP assay**. The following primary antibodies were used in this study: rabbit anti-Actn3 (ab68204) from Abcam; mouse anti-Mib1 (sc-393551), mouse anti-C-Myc (sc-40); mouse anti-HA (sc-7392) and normal mouse IgG (sc-2025) from Santa Cruz; normal rabbit IgG (2729S) and rabbit anti-GapdH (2118S) from Cell Signaling; mouse anti-FLAG (F1804); rabbit anti-β-actin (A2066) from Sigma-Aldrich; horseradish peroxidase (HRP)-conjugated donkey anti-rabbit IgG (711-035-152) and HRP-conjugated donkey anti-mouse IgG (715-035-151) from Jackson; and HRP-conjugated anti-mouse IgG (w4021) and HRP-conjugated anti-rabbit IgG (w4011) from Promega. All primary antibodies were diluted 1:1000 with TBS containing 0.1% Tween-20 and 5% bovine serum albumin (BSA). For immunoblot assay, mouse or human muscle tissues were homogenized and lysed in RIPA buffer supplemented with 1× protease inhibitor cocktail (Thermo Fisher) and 1 μg mL$^{-1}$ pepstatin. Cell lysates were subjected to sodium dodecyl sulfate-polyacrylamide gel electrophoresis (SDS-PAGE), transferred to PVDF membranes (MiliporeSigma), blocked with 5% BSA for 1 h at RT, and immunoblotted with indicated antibodies. For insoluble muscle fraction, muscle tissue was homogenized and lysed in mild detergent buffer containing 1% Triton X-100, 50 mM Tris-HCl pH 8, 100 mM NaCl, 1 mM EDTA, 1× protease inhibitor cocktail (Thermo Fisher), and 1 μg mL$^{-1}$ pepstatin. Lysed muscle tissues were incubated on ice for 1 h and centrifuged at 17,000 × g at 4 °C for 10 min. The remaining pellets were solubilized in 2% SDS for 1 h on ice and centrifuged for 17,000 × g for 30 min. The supernatants (insoluble fractions) were collected and subjected to immunoblot. For the co-IP assay, muscle tissues and cells were harvested and lysed in IP buffer (0.5% or 0.05% Nonidet P-40, 2 mM EDTA, 10% glycerol in PBS) supplemented

with 1× protease inhibitors (Thermo Fisher) and 1 µg mL$^{-1}$ pepstatin. Protein lysates were precleared with protein A/G Plus Agarose beads (Santa) for 1 h at 4 °C, while beads and primary antibody were incubated for 1 h at RT. Primary antibody-conjugated beads were incubated with precleared lysates overnight at 4 °C. The precipitated protein was washed with IP buffer, eluted in 2× Laemmli sample buffer followed by heating at 100 °C for 10 min, and subjected to SDS-PAGE and immunoblotted. The membranes were developed using the FUSION Solo chemiluminescence imaging system (Vilber, Marne-la-Vallée, France). The intensity of protein expression of immunoblot was quantified by ImageJ (1.51k).

**Histochemistry and immunohistochemistry.** Dissected muscle tissue was embedded in the Tissue Teck OCT compound (Sakura Fineteck, Torrance, CA, USA), quickly frozen in liquid nitrogen, and stored at −80 °C prior to sectioning. Muscles were sectioned at 7 µm. Basic muscle morphology was assessed with hematoxylin and eosin or Gomori's Trichrome staining according to standard protocols. For myosin heavy chain (MyHC) staining, unfixed muscle sections were incubated overnight at 4 °C with mouse anti-MyHC type 1 (DSHB, BA-D5, 1:100 dilution), mouse anti-MyHC type 2a (DSHB, SC-71, 1:100 dilution), mouse anti-MyHC type 2b (DSHB, BF-F3, 1:100 dilution), or mouse anti-MyHC type 2x (DSHB, 6H-1, 1:5 dilution) in addition to rat anti-laminin (Abcam, ab11576, 1:2000 dilution) in 5% BSA blocking buffer. After washes in PBS, sections were incubated for 1 h with 1:400 dilution of Alexa Fluor 488-conjugated anti-mouse IgG (Invitrogen), Alexa Fluor 488-conjugated anti-mouse IgM (Invitrogen), Alexa Fluor 594-conjugated anti-rabbit IgG (Invitrogen), and Alexa Fluor 594-conjugated anti-rat IgG (Invitrogen). For muscle stem cell staining, sections were fixed in 4% paraformaldehyde and subjected to epitope retrieval using sodium citrate (0.01 M, pH 6.0) at 105 °C for 10 min. Sections were treated in the M.O.M (mouse on mouse) blocking solution, according to the manufacturer's instructions (FMK-2201, Vector Laboratories, Burlingame, CA, USA) and incubated with mouse anti-Pax7 (DSHB, Pax7, 1:100 dilution) and rabbit anti-laminin (Sigma, L9393, 1:2000 dilution). After washing in PBS, sections were incubated with 1:400 dilution of Alexa Fluor 594-conjugated anti-mouse IgG (Invitrogen) and Alexa Fluor 488-conjugated anti-rabbit IgG (Invitrogen). Nuclei were counterstained with Hoeschst (Invitrogen). Images were visualized with a Zeiss Observer Z1 fluorescent microscope (Zeiss, Oberkochen, Germany) and captured with a Spot Flex camera. The Actn3-stained whole cross-section images were captured with EVOS FL Auto 2 (Thermo Fisher, Waltham, MA, USA) at the same light setting, exposure time, and magnification. For fiber CSA calculation, Leopard (ZOOTOS, Korea) was used. For fiber CSA calculation via Actn3 intensity, semiautomatic muscle analysis using segmentation of histology was used. Briefly, segmentation of histology analysis was used with a segmentation filter (CSA over 250 µm$^2$ and <5000 µm$^2$; eccentricity ≤0.98; convexity ≥0.78) on laminin-stained sections. The average Actn3 intensity per myofiber and corresponding CSA of myofiber were analyzed according to Smith and Barton[77].

**Yeast two-hybrid screening assay.** For bait construction, the full-length mouse Mib1 gene was amplified by PCR and cloned into the pGBKT7 vector containing the GAL4 DNA-binding domain. Bait construct did not show autonomous transcriptional activation and toxic effect for growth following transformation into the yeast strain, Y2H Gold. For library construction, the mouse myofiber cDNA library was constructed in a pGADT7-Rec vector containing a GAL4 activation domain using Matchmaker Library Construction and Screening Kits (Clontech, Kusatsu, Japan) and then transformed into the Y187 yeast strain. Yeast two-hybrid screening was performed using the Matchmaker Two-Hybrid system instruction (Clontech) according to the manufacturer's instructions. Positive clones were selected on synthetic dropout medium/−Leu/−Trp/−His/−Ade/+AbA/+X-α-Gal (QDO/+X/+A) and cDNA inserts were identified by PCR amplification, sequencing, and BLAST alignment. The same positive and negative controls were included. The positive clones among primary transformants were PCR amplified, sequenced for clones over 700 bp PCR products, and aligned using the NCBI BLAST alignment search tool (http://blast.ncbi.nlm.nih.gov/Blast.cgi).

**In vivo ubiquitination assay.** Cells were transfected with indicated plasmids and treated with 10 µM MG-132 (Sigma) for 6 h. Cells were collected and lysed in SDS lysis buffer (5% SDS, 30% glycerol, 150 mM Tris pH 8.0). Lysates were incubated with the anti-C-Myc antibody for 2 h and protein A/G agarose beads for a further 2 h at 4 °C. The precipitated proteins were washed and then eluted in 2× Laemmli sample buffer followed by heating at 100 °C for 10 min, and subjected to SDS-PAGE and immunoblotted.

**Cycloheximide chase assay.** Cells transfected with the indicated plasmids were treated with 40 µM cycloheximide (Sigma) or 0.1% dimethyl sulfoxide (Sigma) for the indicated times before collection.

**Grip strength measurement.** Whole-limb grip strength was assessed by allowing mice to grab a grid that is attached to a grip strength test meter (Grip strength test, Bioseb, France). The randomized mouse is pulled horizontally by the tail away from the grid and the peak force was measured. The maximum absolute grip strength for each mouse was normalized to body mass (g). All experiments were performed in a blind fashion.

**Acute and chronic exercise training.** The randomized mice were pre-acclimated to the treadmill (DJ2-242, Dual Treadmill, Daejeon, Korea) for 3 days by exploration (0 m min$^{-1}$ for 5 min) and subsequent running with speed of 5 m min$^{-1}$ for 5 min, 10 m min$^{-1}$ for 5 min, and 15 m min$^{-1}$ for 5 min. Mice were then subjected to treadmill running with 5 m min$^{-1}$ speed increments every 5 min up to 15 m min$^{-1}$ speed until exhaustion. Running time and distance were recorded and collected for each mouse. Mice were considered to be exhausted when remained at the end of the treadmill for >10 s despite mechanical prodding. The chronic exercise training was performed for 5 days per week for 6 weeks with a speed of 5 m min$^{-1}$ for 5 min, 10 m min$^{-1}$ for 5 min, and 15 m min$^{-1}$ for 30 min. All experiments were performed in a blinded fashion.

**Statistics.** Data are represented as mean ± SEM. Statistical analyses were performed with Prism 5.01 (GraphPad). Comparisons relied on one- or two-way analysis of variance followed by Tukey's multiple-comparisons post hoc test and Pearson's $\chi^2$ test. Otherwise, unpaired two-tailed $t$ tests were performed. $P$ values < 0.05 were considered significant.

**Reporting summary.** Further information on research design is available in the Nature Research Reporting Summary linked to this article.

## Data availability

The authors declare that the data supporting the findings of this study are available within the paper. Other data that support the findings of this study are available from the corresponding author upon reasonable request. Source data are provided with this paper.

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

## Acknowledgements

We thank Drs. Sang-Hee Lee, Bong-Yoon Kim and Jong Ho Park for technical support on TEM analysis yeast two-hybrid screening assay and in vivo ubiquitination assay, respectively. This work was supported by the National Research Foundation of Korea (NRF-2017R1A2B3007797 and NRF-2020R1A5A1018081), Korea Mouse Phenotyping Project (NRF-2014M3A9D5A01073930) of the Ministry of Science and ICT, and National Research Foundation of Korea, and funded by the Korean government (MSIT) and by SNU-Yonsei Research Cooperation Program through Seoul National University.

## Author contributions

J.-Y.S. designed the study, performed the experiments, and wrote the manuscript. J.-S.K., Y.L.K., J.-H.K., Y.W.J., S.-H.H., J.P., I.P., K.Y., J.R., and H.P. conducted the experiments. J.-W.P and Y.C.H. performed hip joint surgery, prepared human skeletal muscle tissue samples, and provided helpful comments. Y.-Y.K. designed the research and wrote the manuscript.

## Competing interests

The authors declare no competing interests.
