## [Peer Review File · Nature Communications]

Reviewers' Comments:

Reviewer #1:

Remarks to the Author:

The manuscript by Seo et al. demonstrates a role of skeletal muscle Mib1 on age-associated muscle atrophy. The authors also show an interesting role for Mib1 in protection against muscle atrophy in young mice subjected to 6 wks of endurance training. While the data regarding Mib1 as a regulator of muscle mass are convincing, there are some aspects of the manuscript that would benefit from clarification or additional evidence.

1. The data regarding Mib1 as a regulator of muscle mass are convincing, however the hypothesis that muscle atrophy is regulated by increases in Actn3 is less convincing. Addressing the following concerns may strengthen this hypothesis:

a. It would be preferable to see the change in Mib1 and Actn3 proteins with age in the same mouse samples. While Figure 1 shows a decrease in Mib1 levels at 26 months, Figure 4 shows an increase in Actn3 at 30 months and no 26 month samples are included. The use of different sets of aged mice to show the changes in these two proteins with age, makes it more difficult to make a case for co-regulation.

b. Data from humans is not convincing, as the blot looks like Actn3 is lower in 'aged' group. This may be because the "loading control" protein (GAPDH) is not consistent among the lanes and also appears to be lower in the aged group. A more appropriate loading control should be selected. In addition, the authors show that low Actn3 is associated with sarcopenia, which seemingly contradicts their hypothesis. While they explain this by saying that a loss of glycolytic fibers may be to blame, there is significant evidence that low Actn3 due to a highly common genetic variant can predict sarcopenia with age. Indeed, a significant proportion of the population does not express functional Actn3, and this has nothing to do with age-associated declines in Mib1, but rather an inherited genotype. Since absence of Actn3 is so common, healthy phenotypes in humans are associated with a wide variety of Actn3 expression, thus it does not appear necessary for normal muscle function, although its presence can impart improved strength and power. This would argue against a role for modest increases in Actn3 causing pronounced muscle dysfunction.

c. The absence of Actn3 is associated with improved endurance performance in knockout models and humans. Thus, the reduction of endurance capacity in Mib1 mice does not seem likely to be due to an effect on Actn3.

d. The EM images in Figure 2 show profound muscle dysregulation and there is a loss of myofibers in Mib1 mice suggesting apoptosis. Is there a hypothesis as to how increased Actn3 would cause such a phenotype? Is it possible that Mib1 can regulate apoptosis through a Mib1-independent mechanism?

e. Showing levels of other common ubiquitin ligases (MuRF/MAFbx) that contribute to age-associated muscle atrophy in Mib1 KO mice would be informative.

f. In Figure 5f, it would be helpful to see the level of ACTN3 in the sedentary samples, since it is unclear if the increase seen in Mib1 KO is due to the exercise, or simply reflects basal levels. If this were the case, the exercise-induced atrophy would be unrelated to the Actn3 levels. Likewise, in Figure 5i, it would be helpful to compare the knockdown of Actn3 with the endogenous levels of Actn3 in WT mice. It is hard to determine whether the Actn3 has been "normalized" without the control group shown.

g. While knockdown of Actn3 appears to prevent some of the muscle atrophy with exercise in Mib1 Knockout mice (Figure 5k), the magnitude of the increase in muscle mass is much smaller than the Mib1-induced decrease. This is despite a very significant decrease in Actn3 protein. These results would suggest that Actn3 is only responsible for a small fraction of the Mib1-associated muscle loss with exercise.

2. MCK promoter preferentially affects expression in glycolytic fibers. Could this be the reason for Type II specific atrophy? What is the regular fiber-type distribution of Mib1?

3. For the ubiquitination experiment in Figure 4b, there appears to be less Actn3 (Myc) in the lane with the inactive version of Mib1. Could this be the reason for the lower ubiquitination signal for the Myc IP? There are also no statistics listed for this experiment. Were the results quantified for independent experiments, and was statistical analysis performed?

4. For the cyclohexamide chase experiment, the levels of Mib1 (FLAG) aren't shown, and each well appears to contain overexpressed Mib1. Wouldn't a control experiment to show the degradation of Actn3 in the absence of Mib1 be necessary to show this effect is specific to Mib1?

Minor Comments:

Figure 1j: are pictures of fat pads reversed? The fat pads pictured for Mib1 KO appear smaller than WT, but the data would suggest the opposite.

Line 181: Should this say "endogenous co-immunoprecipitation"?

Histological features in Figure 2i are difficult to see.

Reviewer #2:

Remarks to the Author:

The study by Seo and collaborators addresses the role that the E3 ubiquitin ligase Mind-bomb (Mib)1 plays in the skeletal muscle homeostasis by regulating the degradation of the actin-binding protein Actn3. In particular, the study refers to the reduced Mib1 expression occurring during aging that results in Actn3 accumulation and, consequently, in muscle degeneration. The study uses different approaches to reach such a conclusion and is rather well designed. There are, however, some points that should be addressed:

- the authors demonstrate that mice lacking Mib1 (Mib1 Δ MF) in the skeletal muscle present with a phenotype that is consistent with accelerated aging, including a reduced number of muscle stem cells. In order to investigate if such lack of stem cells could account for the premature aging features, they study mice in which Pax7+ cells have been genetically deleted (SCDTA) at the same age of Mib1 Δ MF animals. The results show that 16 months old SCDTA mice do not show any of the alterations occurring in Mib1 Δ MF animals. On this basis the authors conclude that the lack of muscle stem cells does not account for accelerated aging in Mib1 Δ MF animals. However, this conclusion is not really supported by the data presented in the study. Indeed, the experiment shown by the authors demonstrates that muscles deprived of stem cells do not conform to the premature aging phenotype. However, they do not clarify the role of low muscle stem cell number in the Mib1 Δ MF background. The possibility that in the Mib1 Δ MF microenvironment the reduced muscle stem cell number could contribute to alter the skeletal muscle homeostasis cannot be discarded. This point could be easily unraveled by enriching Mib1 Δ MF muscles with stem cells, obtained from both wild type and genetically modified mice and testing if the accelerated aging features can still be observed. In addition, there are quite interesting data produced by the group coordinated by Dr. Pura Muñoz Canoves showing that aged muscle stem cells lose their functionality due to increased oxidative stress and reduced ability to activate autophagy. Both these possibility should be investigated, or at least discussed, in the present study;
- the authors show in Figure 1a and 1b that Mib1 levels are reduced in 16 and even more in 26 months old mice compared to 3 months old animals. When does Mib1 starts to decline? Six, nine months of age? Can such decline be causally correlated with other parameters such as oxidative stress and/or inflammaging? In other words, what is causing Mib1 down-regulation?
- the study does not provide any assessment of muscle function;
- what about animal survival? do Mib1 Δ MF mice live less than WT animals?
- is Mib1 physiologically expressed also in oxidative fibers? is muscle phenotype changed by Mib1 modulations (down- or up-regulated) in these fibers?
- how is the Notch-dependent system behaving in the 16 months old Mib1 Δ MF mice?
- are muscle-specific features of the Mib1 Δ MF mice comparable to other models of accelerated aging? Or, taking the problem from the opposite point: are Mib1 levels reduced also in models such as the SAMP8 mouse?
- Mib1 lack is associated with reduced myofiber number (Figure 2f), mainly reflecting myosin heavy chain 2b -expressing fibers (Figure 2g), suggesting a selective effect. This observation stimulates a number of questions: a) does the reduced fiber number depend on altered myogenesis (either due to lack of muscle stem cells or to their reduced functionality)? b) is selective myofiber 'death' occurring in 16 months old Mib1 Δ MF mice? c) if this is the case, which mechanism is driving such selective loss?

- Figure 1h shows that among the three different adipose tissues examined from Mib1 Δ MF mice, the epididymal one is not enlarged compared to WT animals, but it is darker (Figure 1j). Is browning of occurring in Mib1 Δ MF mice, at least at the epididymal level?
- page 16, first line: 'muscle myopathy' is a sort of tautology...

Point-by-point reply in response to the reviewer's comments

Reviewer 1

1. It would be preferable to see the change in Mib1 and Actn3 proteins with age in the same mouse samples. While Figure 1 shows a decrease in Mib1 levels at 26 months, Figure 4 shows an increase in Actn3 at 30 months and no 26 month samples are included. The use of different sets of aged mice to show the changes in these two proteins with age, makes it more difficult to make a case for co-regulation.

- We fully agreed with the reviewer's suggestion that showing the age-associated changes in Mib1 and Actn3 levels in the same mouse would strengthen our data about the Mib1-Actn3 axis in skeletal muscle with age. At the initial stage of our study, we only examined Mib1 expression levels to examine the age-associated change in Mib1. In further experiments to investigate the mechanism by which Mib1 regulates age-associated muscle atrophy, we identified Actn3 as a binding partner of Mib1 and analyzed Actn3 levels with age. As the reviewer pointed out, we separately examined the change in Mib1 and Actn3 protein levels with age. Now we thoroughly re-analyzed the age-dependent Mib1 and Actn3 protein levels in the same mouse samples. As shown in new Figure 1a,b, and Figure 4f,g, Mib1 expression level begins to decline around 16 months of age, whereas Actn3 showed delayed accumulation around 30 months of age, suggesting that Actn3 progressively accumulates as Mib1 level falls below a certain threshold. Consistently, when Mib1 is deficient in myofibers, Actn3 protein levels increased at 16-month-old while remained comparable at 3-month-old (new Extended Data Figure 4c and Figure 4h,i), further indicating that the accumulation of Actn3 gradually occurs by the prolonged absence of Mib1. Now we replaced Fig.1a,b.

New Figure 1a and b. Age-associated expression levels of Mib1 in skeletal muscles.

2. Data from humans is not convincing, as the blot looks like Actn3 is lower in 'aged' group. This may be because the "loading control" protein (GAPDH) is not consistent among the lanes and also appears to be lower in the aged group. A more appropriate loading control should be selected (a). In addition,

the authors show that low Actn3 is associated with sarcopenia, which seemingly contradicts their hypothesis. While they explain this by saying that a loss of glycolytic fibers may be to blame, there is significant evidence that low Actn3 due to a highly common genetic variant can predict sarcopenia with age. Indeed, a significant proportion of the population does not express functional Actn3, and this has nothing to do with age-associated declines in Mib1, but rather an inherited genotype. Since absence of Actn3 is so common, healthy phenotypes in humans are associated with a wide variety of Actn3 expression, thus it does not appear necessary for normal muscle function, although its presence can impart improved strength and power. This would argue against a role for modest increases in Actn3 causing pronounced muscle dysfunction (b).

- (a) As the reviewer pointed out, we apologize for inconsistent GAPDH protein levels in human data due to our technical carelessness. We are also aware of the importance of the appropriate loading control. Although we tried to load the same amount of protein lysates, it was difficult to precisely control the loading amounts of human skeletal muscle tissues. While ACTN3 levels in the Aged group remain similar compared to the Middle-Aged group, we observed lower GAPDH protein levels in aged group (previous Figure 6a). Based on these data, we presumed that this data may highlight an increase in ACTN3 levels in the Aged group compared to other groups. As the reviewer suggested, searching for an appropriate loading control would be a good option. Indeed, we have tried to normalize loading amounts with beta-actin and tubulin (Attached Figure 1). However, these data were even worse than that of GAPDH. During our trials, unfortunately, tissue lysates from most human samples were exhausted, because tissue sizes of human biopsies were not enough to conduct immunoblotting as well as other experiments. As a last attempt, we used GAPDH as the loading control, because we had the more information based on previous trials. We re-loaded lysates, blotted with GAPDH, and re-measured band intensity. Now we replaced Fig. 6a-c.

Attached Figure 1. β -tubulin, β -actin, and GapdH levels with age in human samples

New Figure 6a-c. Mib1 and Actn3 levels with age in human samples.

- (b) We appreciated the reviewer's critical comment on the implication of low Actn3 on sarcopenia. As the reviewer pointed out, several studies reported the implication of Actn3 genotype and sarcopenia¹⁻³. In order to be diagnosed with sarcopenia, the skeletal muscle mass index (SMI) should be lower than 7.0-7.26 or 5.4-5.67kg/m² for men and women, respectively, according to sarcopenic criteria⁴⁻⁷. Even though several studies reported the effects of Actn3 genotypes in aging muscles, the SMIs of both RR (functional Actn3) and XX (non-functional Actn3) genotype were lower or higher than sarcopenic criteria^{1,2}. Moreover, other studies reported no association between Actn3 genotype and aging¹⁰⁻¹³. In addition, a 5-year follow-up study conducted by Delmonico et al., showed that Actn3 genotype was associated with the longitudinal measures of some physical functioning but there was no significant association between Actn3 genotypes and sarcopenia¹². Considering various human studies on Actn3 R577X polymorphism, it still seems controversial whether Actn3 R577X polymorphism is implicated in sarcopenia

Despite the controversy, we conducted SNP genotyping to avoid the implication of Actn3 polymorphism in our study¹⁴. Briefly, we isolated gDNA from human skeletal muscles and performed Actn3 R577X genotyping using the following primers: ACTN3-F 5'-CTGTTGCCTGTGGTAAGTGGG-3' and ACTN3-R 5'-TGGTCACAGTATGCAGGAGGG-3'. As shown in New Extended Data Figure 7b, there was no Actn3 polymorphism (arrows indicate fragments of 205 and 86 bp) in our human samples, indicating that loss of Actn3 in the Sarcopenia group is not associated with inherited genotypes, but rather the loss of Mib1 followed by

accumulation of Actn3 and subsequent loss of selective type 2 glycolytic myofibers with age.

New Extended Data Figure 7b. Actn3 R577X genotyping using gDNA isolated from human skeletal muscles.

Although all human specimens we examined in this study had an Actn3 RR genotype (functional Actn3), the expression of Actn3 was very low only in the Sarcopenia group compared to those of other groups. Actn3 is predominantly expressed in type 2 glycolytic myofiber⁹ and the loss of type 2x glycolytic myofiber (the skeletal muscle of humans is composed of type 1 oxidative, 2a oxidative, and 2x glycolytic myofibers¹⁵) is a hallmark of sarcopenia¹⁶. As reported, the number of type 2x glycolytic myofibers in the sarcopenic patients was greatly decreased in our study (New Figure 6d,e). These data suggest that the loss of Actn3 in the sarcopenic patients in this study is at least not due to Actn3 polymorphism but rather the loss of type 2x glycolytic myofibers. As sarcopenia becomes an increasingly important disease, it might be necessary to unravel how the loss of type 2x glycolytic myofibers occurs in sarcopenic patients. We propose here that our findings in this study suggest at least one of the mechanisms implicated in the loss of type 2 glycolytic myofibers in sarcopenia.

New Figure 6d,e. Quantification of type 2x glycolytic myofibers (d) and representative IHC staining for MyHC2x and Laminin of skeletal muscles (e).

Collectively, we re-loaded lysates and replaced Fig. 6a-c. In addition, we added data (Figure 6d, e, and Extended data Figure 7b) and description (page 13-14, line 303-317).

3. The absence of Actn3 is associated with improved endurance performance in knockout models and humans. Thus, the reduction of endurance capacity in Mib1 mice does not seem likely to be due to an effect on Actn3.

- As the reviewer pointed out, the absence of Actn3 results in the improvement of endurance performance in Actn3 knockout mouse models and humans¹⁷⁻¹⁹ because Actn3 interacts with a broad range of proteins involved in structural, metabolic, and signaling²⁰. Actn3 has been known as a gene of speed because Actn3 genotype is closely associated with skeletal muscle performance. Thus, most studies have been primarily focused on a loss-of-function study rather than a gain-of-function study. In contrast to the absence of Actn3, Garton et al. recently suggested that the high dose of Actn3 in skeletal muscle can be detrimental to muscle function²¹ although they did not give a clear answer to the mechanism by which high level of Actn3 leads to muscle damage and dysfunction. Intriguingly, the skeletal muscle with supraphysiological overexpression of Actn3 showed reduced force generation and altered fatigue recovery²¹, further suggesting the adverse effects of Actn3 overexpression on skeletal muscle function. The possible reasons how supraphysiological accumulation of Actn3 results in the reduction of endurance capacity in Mib1^{ΔMF} mice are as follows:

First, the progressive accumulation of Actn3 over a certain threshold might overburden the variable layers of Z-disks. Z-disks are composed of two to six variable layers that can bear the accumulation of sarcomeric proteins up to a certain level^{21,22} and contraction-induced mechanical stress²³. Thus, as Actn3 gradually accumulated, Z-disks might no longer withstand and subsequently be disorganized, further developing an accumulation of tubular aggregates and aberrant membrane structure. Moreover, several studies reported that the accumulation of protein aggregates and/or sarcomeric protein aggregates leads to failure of maintenance of sarcomeric integrity, and subsequent muscle degeneration and impairment of contractile activity²⁴⁻²⁷. Indeed, Z-disks were disorganized and tubular aggregates were accumulated in Mib1^{ΔMF} mice (Figure 2I-o), indicating that accumulation of Actn3 over certain threshold leads to disorganized sarcomeric structure, resulting in impaired muscle function.

Secondly, the accumulation of Actn3 might result in disturbed assembly of sarcomeric proteins such as titin, desmin, vinculin, and Z-band alternatively spliced PDZ motif-containing protein (ZASP) that are known to bind with Actn3¹⁹. Among these proteins, titin and desmin are one of the major Z-disk proteins involved in sarcomere assembly, mechanosensor, and force transducer function²⁸. The disorganized Z-disk in Mib1^{ΔMF} mice suggests the possibility of impaired muscle contractile activity. For example, mutated desmin adversely affects the endurance

function of mice because of low force generation and transmission²⁹. In addition, when titin is impaired, the force generation and transmission are also impaired since titin is an elastic element which contributes to passive tension on a force-length relationship in myofibers³⁰. Considering the interaction between Actn3 and other sarcomeric proteins in Z-disks, the accumulation of Actn3 might lead to abnormal assembly of sarcomeric proteins which are essential for force generation and transmission.

Collectively, we added the description in the discussion regarding this issue (page 15, line 345-348 and 352-353; page 16, line 373-374 and 377-382; page 16-17, line 377-393).

4. The EM images in Figure 2 show profound muscle dysregulation and there is a loss of myofibers in Mib1 mice suggesting apoptosis. Is there a hypothesis as to how increased Actn3 would cause such a phenotype? (a). Is it possible that Mib1 can regulate apoptosis through a Mib1-independent mechanism? (b).

- (a) We appreciated the reviewer's comment. We also think that the loss of myofibers in Mib1^{ΔMF} mice is mediated by apoptosis. Our study and Garton et al., showed that the excessive accumulation of Actn3 over a certain threshold in Z-disks results in detrimental effects on skeletal muscle and skeletal muscle atrophy²¹. Actn3 forms antiparallel homodimers that cross-link actin filaments and also interact with titin in the Z-disks. The dimeric Actn3 exists as a closed conformation. However, upon activation, it changes its conformation to an open form and binds to other sarcomeric proteins³¹. The dimerization has a high propensity to form protein aggregation, resulting in toxic aggregates that can cause cytotoxicity or proteotoxicity³²⁻³⁵. Thus, increased concentration of dimeric Actn3 can act as seeds for further aggregation. In Figure 4h-i and 5f, we showed the accumulation of Actn3 in total muscle lysates and an insoluble fraction of muscle lysates, respectively, indicating that increase in Actn3 protein concentration can lead to the formation of large insoluble aggregates in skeletal muscle. As reported, the accumulation of protein aggregates such as dimeric Actn3 aggregates in skeletal muscle can lead to toxic accumulation and subsequently proteotoxic stress which can activate apoptotic pathway³⁶⁻³⁹.

- (b) We presumed that the reviewer questioned whether Mib1 can regulate apoptosis through a Mib1-Notch-independent mechanism. In this study, we unravel the novel function of Mib1 that regulates Actn3 via proteasomal degradation pathway in skeletal muscle. In Mib1-deficient myofibers, Actn3 accumulates over a certain threshold, leading to toxic accumulation and possibly subsequent proteotoxic stress

that activates the apoptotic pathway. Although the loss of Mib1 leads to muscle atrophy, it does not imply that Mib1 directly regulates apoptosis in myofibers because apoptosis is activated by proteotoxic stress induced by excessive accumulation of proteins. As aforementioned, apoptosis would be a byproduct of Actn3 accumulation, and thus the Mib1-Notch pathway would not be the mediator of apoptosis.

Collectively, we added the description in the discussion regarding this issue (page 17-18, line 402-419).

5. Showing levels of other common ubiquitin ligases (MuRF/MAFbx) that contribute to age-associated muscle atrophy in Mib1 KO mice would be informative.

- We thank the reviewer for highlighting this point. As the reviewer suggested, we examined the levels of other common E3 ubiquitin ligases (MuRF and MAFbx/Atrogin1) in 16-month-old Mib1^{WT} and Mib1^{ΔMF} GA muscles. As shown in the figure below, there was no change in the expression levels of E3 ubiquitin ligases, further indicating that muscle atrophy in 16-month-old Mib1^{ΔMF} mice is solely due to loss of E3 ubiquitin ligase Mib1. Now we added Extended Data Figure 3g and description in the result regarding this issue (Page 9-10, line 210-212, and 215-218)

New Extended Data Figure 3g. Relative mRNA expression levels of E3 ubiquitin ligases associated with muscle atrophy in Mib1^{WT} and Mib1^{ΔMF} mice

6. In Figure 5f, it would be helpful to see the level of ACTN3 in the sedentary samples, since it is unclear if the increase seen in Mib1 KO is due to the exercise, or simply reflects basal levels. If this were the case, the exercise-induced atrophy would be unrelated to the Actn3 levels (a). Likewise, in Figure

5i, it would be helpful to compare the knockdown of Actn3 with the endogenous levels of Actn3 in WT mice. It is hard to determine whether the Actn3 has been “normalized” without the control group shown (b).

- (a) We apologize for insufficient data on the basal level of Actn3 in sedentary young mice due to space limitation. We examined the Actn3 levels at sedentary 3-month-old Mib1^{WT} and Mib1^{ΔMF} mice and found that the Actn3 levels were comparable between groups (Extended Data Figure 3c). When we chronically exercised 3-month-old Mib1^{WT} and Mib1^{ΔMF} mice for 6 weeks, Actn3 levels were comparable between groups until 2 weeks of chronic exercise (Fig. 5f,g). Importantly, Actn3 levels were not increased in Mib1^{WT} mice 6 weeks after chronic exercise, whereas markedly increased in chronic exercised Mib1^{ΔMF} mice, indicating that increase of Actn3 in the Mib1^{ΔMF} mice is due to chronic exercise. Now, we added western blot data to show the comparable endogenous Actn3 levels in sedentary 3-month-old Mib1^{WT} and Mib1^{ΔMF} mice (New Extended Data Figure 4c).

New Extended Data Figure 4c. Actn3 levels in sedentary 3-month-old Mib1^{WT} and Mib1^{ΔMF} mice.

- (b) We fully agreed with the reviewer’s opinion that it is important to determine the appropriate control to judge whether the experimental objectives are valid. The objectives of Figure 5h-n are that decreased CSA and the number of type 2b myofibers in the exercised Mib1^{ΔMF} mice are caused by increased Actn3. To investigate whether increased Actn3 is responsible for the decreased CSA and number of type 2b myofibers in the exercised Mib1^{ΔMF} mice, we delivered shActn3 into the exercised Mib1^{ΔMF} mice to reduce the accumulation of Actn3 in Mib1^{ΔMF} mice. We believe that shCtrl-delivered exercised Mib1^{ΔMF} mice that still have increased Actn3 should be the appropriate control to estimate whether Actn3 is involved in the decreased CSA and number of type 2b myofibers in the exercised Mib1^{ΔMF} mice. As the reviewer pointed out, the selection of a control group is essential to minimize the effect of all variables besides the independent variable of the experiment. In this case, Actn3 levels are the only independent variable that can evaluate its effect.

Although the reviewer suggested shActn3-delivered Mib1^{WT} mice as a control, Mib1^{WT} showed sustained Actn3 levels after chronic moderate exercise (Figure 5f) and no changes in muscle masses (Extended Data Fig.4 j-l) as well as CSA and number of type 2b myofibers (Data not shown). Thus, we believe that it might not be

helpful to determine the *bona fide* downregulation effects of Actn3 on chronic moderate exercise-induced muscle atrophy in Mib1^{ΔMF} mice. As described in response to comment #7, there was an internal control within shActn3-injected Mib1^{ΔMF} mice due to the uneven delivery of lentivirus (Please see below).

Collectively, we added Extended Data Figure 4c and description regarding this issue (page 11, line 240-241; page 11, line 261-262).

7. While knockdown of Actn3 appears to prevent some of the muscle atrophy with exercise in Mib1 Knockout mice (Figure 5k), the magnitude of the increase in muscle mass is much smaller than the Mib1-induced decrease. This is despite a very significant decrease in Actn3 protein. These results would suggest that Actn3 is only responsible for a small fraction of the Mib1-associated muscle loss with exercise.

- As the reviewer pointed out, one of the limitations of this study was the delivery of lentivirus into skeletal muscles. We tried to inject shRNA intramuscularly to spread as evenly as possible, as shown in the schematic view in Fig. 5h. However, as shown in Figure 5l, not all Actn3 was downregulated in *Quadriceps* muscles. Especially, the inner part of the *Quadriceps* muscles of 3-month-old ΔMF-EX+shActn3 mice expressed the low-intensity of Actn3 while the outer part of muscles still expressed the high-intensity of Actn3 (New Figure 5l). This uneven delivery of lentivirus by technical limitation would mask the downregulation effect of Actn3 in whole muscle analysis of Mib1^{ΔMF} mice. Thus, to further investigate the downregulation effect of Actn3 in ΔMF-EX+shActn3 mice, we divided IHC images stained with anti-Actn3 antibody into two parts: the inner part and the outer part which has low and high intensity of Actn3 in ΔMF-EX+shActn3 Q muscles, respectively, using semi-automatic muscle analysis using segmentation of histology (SMASH) program⁴⁰ (Figure 5l-n). If Actn3 is not responsible for the exercise-induced muscle atrophy in Mib1^{ΔMF} mice, we speculated that the CSA of myofibers with low intensity Actn3 would be similar to that of corresponding CSA.

As shown in New Figure 5n, the inner part of ΔMF-EX+shActn3 (shActn3ⁱⁿⁿ) myofibers were significantly larger than the corresponding part of ΔMF-EX+shCtrl (shCtrl^{inner}) [$1470.97 \pm 14.01 \mu\text{m}^2$ (shCtrl^{inner}) vs $1911.01 \pm 16.03 \mu\text{m}^2$ (shActn3ⁱⁿⁿ)], while similar in the outer part [$1969.83 \pm 16.03 \mu\text{m}^2$ (shCtrl^{outer}) vs $1984.29 \pm 21.17 \mu\text{m}^2$ (shActn3^{outer})]. Consequently, these data indicate that Actn3 is responsible for the Mib1-associated muscle loss with chronic moderate exercise. Thus, the technical limitation of lentiviral delivery masked the effects of Actn3 on exercise-induced muscle atrophy in the Mib1^{ΔMF} mice in terms of whole muscle mass. Now we added New Figure 5l-n and description (page 12, line 274-284).

New Figure 5l-m. Representative images of IHC staining for Actn3 (l), and Actn3 fluorescent intensity (m) and CSA (n) of outer and inner parts of $\Delta MF-EX+shCtrl$ and $\Delta MF-EX+shActn3$ Q muscles.

8. MCK promoter preferentially affects expression in glycolytic fibers. Could this be the reason for Type II specific atrophy? (a). What is the regular fiber-type distribution of Mib1? (b).

- (a) The reviewer suggested the possibility of the selective expression of MCK promoter in glycolytic myofibers. Indeed, several studies have reported preferential expression of the MCK promoter in glycolytic myofibers^{41,42}. However, MCK-Cre Tg mice used in this study were generated by using a 6.5 kb genomic DNA fragment from the *Mck* gene containing 1kb modulatory region (MR) 1 which resides within the +740 to +1,721 portion of the *Mck*'s first intron⁴³. MR1 is critical for *Mck* expression in slow- and intermediate- myofibers and thus the MCK promoter without MR1 preferentially express in fast glycolytic myofibers⁴⁴. As aforementioned, because we used MCK-Cre Tg mice which have the MR1, the MCK promoter expresses in all types of myofibers. Indeed, in the previous report^{43,45-47}, Cre recombinase was active in both type 1 and 2 fibers, indicating MCK promoter is expressed well in all types of myofibers.

As described below in (b), Mib1 is expressed in all skeletal muscles regardless of the composition of fiber types (Attached figure 3) and disrupted in all hindlimb muscles which are composed of mixed glycolytic- and oxidative- myofiber, glycolytic myofibers, and oxidative myofibers (Attached Figure 2). In our previous study, we also used the same MCK-Cre transgenic mice to disrupt the *Mib1* gene in myofibers⁴⁸ and reported that *Mib1* is required for the conversion of cycling muscle stem cells into quiescent muscle stem cells at puberty by activating Notch signaling.

In general, the muscle stem cells are found in all hindlimb muscles and the population of muscle stem cells is even higher in type 1 slow myofibers than type 2 fast myofibers⁴⁹. If the MCK promoter is preferentially expressed in type 2 fast glycolytic myofibers, the adult muscle stem cells should be present in all hindlimb muscles composed of mixed glycolytic-and oxidative-myofibers or oxidative myofibers. However, in the previous report, we thoroughly showed the depletion of muscle stem cell populations in all hindlimb muscles of *Mib1*^{ΔMF} mice using immunohistochemical- and fluorescence-activated cell sorting (FACS)- analyses, indicating that the MCK promoter is well expressed in myofibers regardless of myofiber type.

Attached Figure 2. *Mib1* expression levels in hindlimbs of *Mib1*^{WT} and *Mib1*^{ΔMF} mice

- (b) We appreciate the reviewer for highlighting this point. In order to examine the distribution of *Mib1* in fiber types, we immunoblotted all skeletal muscles of 3-month-old wild-type mouse such as mixed glycolytic- and oxidative- fiber types (*Tibialis anterior* (TA), *Gastrocnemius* (GA) and *Quadriceps* muscles), glycolytic myofibers (*Extensor digitorum longus* (EDL)) and oxidative myofibers (*Soleus* (SOL)) against *Mib1*. As shown below (Attached Figure 3), *Mib1* is well distributed in skeletal muscle regardless of fiber types.

Attached Figure 3. *Mib1* expression levels in hindlimb muscles

9. For the ubiquitination experiment in Figure 4b, there appears to be less Actn3 (Myc) in the lane with the inactive version of *Mib1*. Could this be the reason for the lower ubiquitination signal for the Myc IP? (a). There are also no statistics listed for this experiment. Were the results quantified for independent experiments, and was statistical analysis performed? (b).

- (a) According to the reviewer's comment, we repeated the ubiquitination experiment. As in New Figure 4b, a lower ubiquitination signal was observed in

Mib1 Δ RING in presence of similar levels of Actn3 (Myc) compared to that of Mib1WT. We now replaced Figure 4b.

New Figure 4b. IB analysis of ubiquitination and proteasomal degradation of Actn3 in 293T cells.

- (b) We conducted 3 independent experiments for ubiquitination. We quantified ubiquitination signals above the IgG heavy chain (70kDa) signal as we used A/G agarose beads for immunoprecipitation. The intensities of HA (Ub) protein of lane 4 and 5 were 1.46 ± 0.12 and 0.97 ± 0.07 , respectively. For statistical analysis, 2-tailed Student's *t*-test was performed and *p*-value was $p=0.03$. Now we added the data in New Figure 4b.

10. For the cyclohexamide chase experiment, the levels of Mib1 (FLAG) aren't shown, and each well appears to contain overexpressed Mib1. Wouldn't a control experiment to show the degradation of Actn3 in the absence of Mib1 be necessary to show this effect is specific to Mib1?

- According to the reviewer's comment on the cycloheximide chase experiment, we observed decreased Mib1 (FLAG) level with time because cycloheximide inhibits the synthesis of both Mib1 and Actn3 levels (Attached Figure 4).

Attached Figure 4. Cycloheximide chase experiment.

We further conducted the cycloheximide chase experiment without overexpression of Mib1-2XFLAG. As expected, the degradation of Actn3 was far slower without overexpression of Mib1-2XFLAG (New Figure 4e) than that with overexpression of Mib1-2XFLAG (Figure 4d), supporting that Actn3 is degraded in a Mib1- and proteasome-dependent manner.

New Figure 4e. Cycloheximide chase experiment without Mib1-2XFLAG overexpression at indicated time.

Now we added the New Figure 4e and description (page 9, line 206-207).

Minor Comments:

1. Figure 1j: are pictures of fat pads reversed? The fat pads pictured for Mib1 KO appear smaller than WT, but the data would suggest the opposite.

- We apologize for the confusion in Figure 1j. In order to show the pictures of fat pads and skeletal muscles within a rectangular format, we decreased the size of the fat pads of Mib1^{ΔMF} mice and instead put the scale bar to represent the degree of low

magnification. However, to avoid any confusion on the size of the figures of fat pads, we resized the pictures with the same magnification. We resized and replaced Figure 1j into new Figure 1k.

New Figure 1k. Gross morphology of adipose tissues of 16-month-old Mib1^{WT} and Mib1^{ΔMF} mice

2. Line 181: Should this say “endogenous co-immunoprecipitation”?

- We deeply apologize for our mistake. We have corrected the sentence (Page 8, Line 187).

3. Histological features in Figure 2i are difficult to see.

- According to the reviewer’s comment on the quality of our representative data, we replaced Figure 2i.

Reviewer #2 (Remarks to the Author):

1. The authors demonstrate that mice lacking Mib1 (Mib1 Δ MF) in the skeletal muscle present with a phenotype that is consistent with accelerated aging, including a reduced number of muscle stem cells. In order to investigate if such lack of stem cells could account for the premature aging features, they study mice in which Pax7+ cells have been genetically deleted (SCDTA) at the same age of Mib1 Δ MF animals. The results show that 16 months old SCDTA mice do not show any of the alterations occurring in Mib1 Δ MF animals. On this basis the authors conclude that the lack of muscle stem cells does not account for accelerated aging in Mib1 Δ MF animals. However, this conclusion is not really supported by the data presented in the study. Indeed, the experiment shown by the authors demonstrates that muscles deprived of stem

cells do not conform to the premature aging phenotype. However, they do not clarify the role of low muscle stem cell number in the Mib1 Δ MF background. The possibility that in the Mib1 Δ MF microenvironment the reduced muscle stem cell number could contribute to alter the skeletal muscle homeostasis cannot be discarded. This point could be easily unraveled by enriching Mib1 Δ MF muscles with stem cells, obtained from both wild type and genetically modified mice and testing if the accelerated aging features can still be observed (a). In addition, there are quite interesting data produced by the group coordinated by Dr. Pura Muñoz Canoves showing that aged muscle stem cells lose their functionality due to increased oxidative stress and reduced ability to activate autophagy. Both these possibility should be investigated, or at least discussed, in the present study (b).

- (a) We apologize for the insufficient description of the complete absence of muscle stem cells in adult Mib1 Δ MF mice. Because of space limitation, we could not include detailed descriptions about (1) the absence of muscle stem cells in Mib1 Δ MF mice and (2) a dispensable role of muscle stem cells in age-associated muscle atrophy, but instead, we had cited references in our result section.

We previously reported that muscle stem cells are completely depleted in adult Mib1 Δ MF mice due to defective Notch signaling⁴⁸, which is essential to establish a reserve pool of quiescent adult muscle stem cells (referenced in our result section, page 5 line 109; Please see Kim et al. (2016)⁴⁸ for a more detailed study on the mechanism of depletion of muscle stem cells in Mib1 Δ MF mice). Briefly, we showed that sex hormones induce Mib1 expression in myofibers at puberty, which activates Notch signaling in cycling juvenile muscle stem cells and causes them to be converted into adult quiescent muscle stem cells, thereby establishing a reserve pool for a lifetime. Hence, when Mib1 is deficient in myofiber, the conversion of proliferating muscle stem cells into quiescent adult muscle stem cells does not occur, leading to precocious depletion of muscle stem cells. Thus, we hope the reviewer understands that there is no way to obtain Mib1 Δ MF muscles with stem cells. Although Mib1 Δ MF mice have no quiescent muscle stem cells for a lifetime, they have normal cycling juvenile muscle stem cells that undergo myogenic differentiation and maturation during the postnatal period. Hence, 3-month-old Mib1 Δ MF mice have normal muscle architecture as shown in Extended Data Figure 4d-i.

In addition, since muscle stem cells are indispensable for muscle regeneration^{50,51}, it has long been believed that the loss of muscle stem cells with age might cause age-associated muscle atrophy. However, unlike general belief, Fry et al. reported that the depletion of muscle stem cells does not accelerate or exacerbate age-associated muscle atrophy (referenced in our result section, page 5 line 114). Aged muscle

stem cell-depleted (SC^{DTA}) mice showed comparable muscle size, fiber type composition, and muscle function compared to wild-type aged mice⁵². In accordance with Fry et al., our study also showed that the loss of muscle stem cells for prolonged time does not contribute to age-associated muscle atrophy (Extended Data Figure 1). Accordingly, if reduced or depleted muscle stem cells contribute to altering the skeletal muscle homeostasis, 16-month-old or aged SC^{DTA} mice should show any aging phenotypes similar to 16-month-old Mib1^{ΔMF} mice. However, no aging phenotypes were observed in SC^{DTA} mice indicating that the age-associated muscle phenotype observed in Mib1^{ΔMF} mice is irrespective of muscle stem cells.

- (b) As the reviewer suggested, the group coordinated by Dr. Pura Muñoz Canoves showed the decline of muscle stem cell regenerative function with age due to entry into senescence⁵³ and impaired autophagy⁵⁴. As aforementioned, muscle stem cells play an essential role in muscle regeneration but not age-associated muscle atrophy. Maintenance of functional muscle stem cells throughout life is crucial for muscle regeneration. However, with age, muscle stem cells become senescent and depleted due to impaired autophagy resulting in regenerative failure. Although the numbers of muscle stem cell decrease with age and lose regenerative capacity, they are dispensable for age-associated muscle atrophy since prolonged depletion of muscle stem cells does not accelerate or exacerbate age-associated muscle atrophy.

Collectively, we have added the description regarding these issues in the discussion (page 20-21, line 465-491).

2. The authors show in Figure 1a and 1b that Mib1 levels are reduced in 16 and even more in 26 months old mice compared to 3 months old animals. When does Mib1 starts to decline? Six, nine months of age? (a). Can such decline be causally correlated with other parameters such as oxidative stress and/or inflammaging? In other words, what is causing Mib1 down-regulation? (b).

- (a) Following the reviewer's suggestion, we analyzed the expression levels of Mib1 at 3, 6, 9, 12, 17, 22, and 30 months of age (Attached Figure 1). Mib1 expression levels seem to decline around 12-17 months of age, coinciding with the onset of the decrease in sex hormones^{55,56} (Further explanation on the relationship between Mib1 and sex hormones are discussed in (b)).

Attached Figure 1. Age-associated expression levels of Mib1 in skeletal muscles.

- (b) As the reviewer suggested, it would be interesting to examine the correlation between a decrease in Mib1 expression with age and other parameters such as oxidative stress and/or inflammaging. Previously, we reported that mouse *Mib1* promoter has putative AR- and ER-binding sites at positions -244 and -1559, respectively, and thus, is induced in myofibers by sex hormones at puberty. Moreover, Mib1 induction is decreased in 3-month-old Gonadotrophin-releasing hormone-deficient hypogonadal (*Gnrh1^{hpg/hpg}*) mice, in which sex hormones are reduced^{48,57}. Since sex hormones decrease with age, we suspect that a decrease in sex hormones with age might result in a decrease in Mib1 expression, an increase in Actn3 expression, and subsequent acceleration of muscle atrophy (please see page 16, line 309-387). Further studies on the implication of sex hormones and Mib1 in sarcopenia using pharmacologically- or genetically-modified mice are required to reveal a putative correlation between sex hormone and Mib1 with age. Collectively, we added the description in the discussion section (page 20, line 461-463).

3. The study does not provide any assessment of muscle function;

- We presumed that the reviewer questioned about the *ex vivo* assessments such as contractility, fatigability, and/or maximal force. In order to precisely analyze the effects of excessive accumulation of Actn3 solely on muscle function, the isolation of single myofiber with and without excessive expression of Actn3 should be performed because each single myofiber would have different levels of Actn3. It would be great if we can isolate single myofibers according to the expression levels of Actn3 like the isolation of muscle stem cells via fluorescence-activated cell sorting (FACS) according to fluorescence intensity. However, as far as we know, there is no technique suitable for the isolation of single myofiber with certain expression levels of protein. Thus, we could not conduct any *ex vivo* assessment but rather performed *in vivo* longitudinal assessment such as whole-limb grip strength test and treadmill running exhaustion test with age. We truly hope the reviewer understand our technical limitation.

4. What about animal survival? do Mib1 Δ MF mice live less than WT animals?

- We appreciate the reviewer's comment. It would be great if we measure the survival rate of Mib1 Δ MF mice compared to Mib1^{WT} mice. As the reviewer's concern, we were also very curious about the life span of Mib1 Δ MF mice and thus monitored 5 Mib1 Δ MF mice with littermate controls as a pilot study. Although we could not get sufficient numbers for statistics, few Mib1 Δ MF mice (22-26 months of age) showed more frailty phenotype as well as age-associated muscle atrophy compared to age-matched littermate Mib1^{WT} mice (Attached Figure 2). Although we believe that

Mib1^{ΔMF} mice show age-related phenotype earlier than Mib1^{WT} mice, we could not include these data because of small sample numbers. Moreover, because of the space limitation of our animal facility, we could not thoroughly examine the survival rates. We hope the reviewer understand this limitation.

Attached Figure 2. Aged Mib1^{WT} and Mib1^{ΔMF} mice. Gross appearance (a), body weight (b), gross muscle morphology (c), and muscle masses (d) of Aged Mib1^{WT} and Mib1^{ΔMF} mice.

5. Is Mib1 physiologically expressed also in oxidative fibers? (a). Is muscle phenotype changed by Mib1 modulations (down- or up-regulated) in these fibers? (b).

- (a) We appreciate the reviewer for highlighting this point. We immunoblotted various skeletal muscles from a 3-month-old wild-type mouse for Mib1 expression. As shown below (Attached Figure 3), Mib1 is expressed well in all muscles examined;

Tibialis anterior (TA), *Gastrocnemius* (GA), and *Quadriceps* muscles that contain mixed (glycolytic and oxidative) myofibers, *Extensor digitorum longus* (EDL) muscle that dominantly contains glycolytic myofibers and *Soleus* (SOL) muscles that dominantly contains oxidative myofibers.

Attached Figure 3. Mib1 expression levels in hindlimb muscles

- (b) Although Mib1 is expressed in all types of myofibers, the numbers and/or cross-sectional area of oxidative myofibers were comparable between Mib1^{WT} and Mib1^{ΔMF} mice (Figure 2e, f and Extended Data Figure 4h,i). Since Actn3 has been known to be predominantly expressed in type 2 glycolytic myofibers⁹, it is likely that the Mib1-Actn3 axis is implicated in the selective alteration of type 2 glycolytic myofibers. Thus, Mib1 modulation would not change the characteristics of oxidative myofibers.

Collectively, we added the description in the result section (page 6, line 125-128).

6. How is the Notch-dependent system behaving in the 16 months old Mib1ΔMF mice?

- As the reviewer would be well aware of, a myofiber with multiple muscle stem cells is surrounded by a basal lamina, and Notch signaling is a signaling mechanism between adjacent cells. As described in response to comment 1 (a) and 5, there are no muscle stem cells in Mib1^{ΔMF} myofibers, in which the myofiber is the only cell in each myofiber. Thus, Notch signal does not work in Mib1^{ΔMF} myofibers. Furthermore, myofiber and muscle stem cells are known as Notch-sending and receiving cells in skeletal muscle⁴⁸, respectively. Thus, Mib1-deficient myofiber and/or loss of muscle stem cells result in the inactivation of Notch signaling.

Notch ligands such as Dll1, Dll4, Jag1, and Jag2 are well-known substrates of Mib1^{58,59}. Thus, as the reviewer pointed out, Notch signaling in muscle stem cells, adjacent to myofibers, should be affected in Mib1^{ΔMF} mice. To exclude the possible implication of Notch signaling in muscle stem cells, we specifically inactivated Notch signaling in muscle stem cells by crossing *Rbpj^{κ1/f}* and *Notch1^{f/f};Notch2^{f/f}* mice with *Pax7-CreERT* transgenic mice (hereafter Notch^{ASC}). As shown in the figure below, the muscle masses of 16-month-old Notch^{ASC} mice were comparable to Notch^{WT} (New Extended Data Figure 2. a-d). In addition, the muscle function of 16-month-old

Notch^{ASC} mice showed normal muscle function (New Extended Data Figure 2.e-f), suggesting that sustained loss of Mib1 in myofiber leads to muscle atrophy and impaired muscle function with age irrespective of Notch signaling. Now we added the New Extended Data Figure 2 and description (page 5, line 108-119 and page 8, line 171-172).

New Extended Data Figure 2. Age-associated muscle atrophy and impaired muscle function occur in Mib1^{ΔMF} mice is irrespective of Notch signaling. (Please see Extended Data Figure Legend for detailed descriptions).

7. Are muscle-specific features of the Mib1^{ΔMF} mice comparable to other models of accelerated aging? Or, taking the problem from the opposite point: are Mib1 levels reduced also in models such as the SAMP8 mouse?

- Some aging mouse models showed comparable age-associated features of skeletal muscle such as age-associated muscle atrophy, impaired muscle function, abnormal skeletal muscle features⁶⁰ similar to Mib1^{ΔMF} mice. Among aging mouse models, SAMP8 mouse showed tubular aggregates⁶¹, muscle atrophy, and impaired

muscle function⁶², representing similar aging phenotypes of Mib1^{ΔMF} mice at least in skeletal muscles. Interestingly, according to several studies, the SAMP8 mouse showed age-related hypogonadotropic hypogonadism⁶³⁻⁶⁵. Considering our previous report that Mib1 is induced in myofibers by sex hormones⁴⁸, it could be possible that the Mib1 level is reduced in SAMP8 mouse models. Importantly, although SAMP8 and Mib1^{ΔMF} mice showed phenotypes of age-related muscle atrophy, SAMP8 is different from Mib1^{ΔMF} mice. First, Mib1^{ΔMF} mice showed skeletal muscle-specific aging phenotypes while SAMP8 mice showed systemic aging phenotypes such as immune dysfunction, behavioral impairments, brain atrophy, and osteoporosis⁶⁶. Moreover, in case of skeletal muscle, SAMP8 mice showed senescent MuSCs, resulting in failure of muscle regeneration⁵⁴ (Please note that MuSCs are dispensable for sarcopenia as aforementioned in Reviewer's comment #1). SAMP8 also showed increased levels of E3 ubiquitin ligases related to muscle atrophy (MuRF, Atrogin) unlike Mib1^{ΔMF} mice which showed comparable mRNA expression of MuRF and Atrogin compared to Mib1^{WT} mice (Extended Data Figure 3g). In addition, albeit both SAMP8 and Mib1^{ΔMF} mice showed similar aging phenotypes related to skeletal muscle, the causality of age-associated phenotypes in SAMP8 and Mib1^{ΔMF} mice are different because the aging phenotypes of Mib1^{ΔMF} mice are originated solely from the loss of Mib1 in myofibers.

New Extended Data Figure 3g. Relative expression levels of E3 ubiquitin ligases associated with muscle atrophy in Mib1^{WT} and Mib1^{ΔMF} mice

Now we added New Extended Data Figure 3g and description in discussion (page 22, line 504-509).

8. Mib1 lack is associated with reduced myofiber number (Figure 2f), mainly reflecting myosin heavy chain 2b-expressing fibers (Figure 2g), suggesting a selective effect. This observation stimulates a number of questions: does the reduced fiber number depend on altered myogenesis (either due to lack of

muscle stem cells or to their reduced functionality)? (a). Is selective myofiber 'death' occurring in 16 months old Mib1 Δ MF mice? if this is the case, which mechanism is driving such selective loss? (b).

- (a) As described in response to comments 1(a), 5, and 6, adult Mib1 Δ MF mice lack muscle stem cells. Although the Mib1 Δ MF mice have no quiescent muscle stem cells for a lifetime, they have normal cycling juvenile muscle stem cells that undergo myogenic differentiation and maturation during the postnatal period⁴⁸. Thus, sedentary 3-month-old Mib1 Δ MF mice have comparable muscle mass, myofiber size, and myofiber type composition compared to those of Mib1^{WT} mice (Fig. 1d-f and Extended Data Fig. 4d-i). The selective loss of type 2b glycolytic myofibers in 16-month-old Mib1 Δ MF mice is not due to defective myogenesis, since myogenesis in Mib1 Δ MF mice normally occurs during postnatal growth⁴⁸. We believe that the selective loss of type 2b glycolytic myofibers at 16-month-old or after chronically exercised 3-month-old Mib1 Δ MF mice occur in a myofiber-intrinsic manner such as Mib1-mediated progressive accumulation of Actn3 in skeletal muscle for a prolonged time.

- (b) Actn3 is predominantly expressed in type 2 glycolytic myofibers⁹. We believe that the selective loss of myofibers in Mib1 Δ MF mice is mediated by proteotoxic stress-induced apoptosis. Our study and Garton et al., showed that excessive accumulation of Actn3 over a certain threshold in Z-disks results in detrimental effects on skeletal muscle and skeletal muscle atrophy²¹. Actn3 forms antiparallel homodimers that cross-link actin filaments and also interact with titin in the Z-disks³¹. The dimeric Actn3 exists as a closed conformation but upon activation, it changes its conformation to an open form and binds to other sarcomeric proteins³¹. The dimerization has a high propensity to form protein aggregation, resulting in toxic aggregates that can cause cytotoxicity or proteotoxicity³²⁻³⁴. Thus, increased concentration of dimeric Actn3 can act as seeds for further aggregation. In Figure 4h-i and 5f, we showed the accumulation of Actn3 in total muscle lysates and the insoluble fraction of muscle lysates, respectively, indicating that increase in Actn3 protein concentration can lead to the formation of large insoluble aggregates in skeletal muscle. As reported, the accumulation of protein aggregates such as dimeric Actn3 aggregates in skeletal muscle can lead to toxic accumulation and subsequently proteotoxic stress which can activate apoptotic pathway³⁶⁻³⁹.

Collectively, we added the description in the discussion section (page 17-18, line 402-419).

9. Figure 1h shows that among the three different adipose tissues examined from Mib1 Δ MF mice, the epididymal one is not enlarged compared to WT animals, but it is darker (Figure 1j). Is browning of occurring in Mib1 Δ MF mice, at least at the epididymal level?

- The reviewer commented on the browning of epididymal adipose tissues of Mib1 Δ MF mice. In Figure 1j, the color of the epididymal adipose tissues of Mib1 WT seems darker than that of Mib1 Δ MF mice. However, when we examined the adipose tissues of littermate 16-month-old Mib1 WT and Mib1 Δ MF mice, epididymal adipose tissue color doesn't depend on the genotype. As shown in Attached Figure 4, the colors of epididymal adipose tissues are dark and/or light in all Mib1 WT and Mib1 Δ MF mice, indicating darker color of epididymal adipose tissues in Mib1 WT or Mib1 Δ MF mice might be due to aging of epididymal adipose tissues rather than Mib1 genotype. Moreover, according to several studies, epididymal adipose tissue has less browning capacity compared to other white adipose tissues⁶⁷ and doesn't undergo white-to-brown transitioning with age⁶⁸. Hence, it seems browning is not occurring in the epididymal adipose tissues of Mib1 Δ MF mice.

Attached Figure 4. Gross morphology of inguinal (Ing.), epididymal (Epi.) and visceral (Vis.) fat of 16-month-old littermate Mib1 WT and Mib1 Δ MF mice.

10. page 16, first line: 'muscle myopathy' is a sort of tautology...

- According to reviewer's comment, we changed the term "muscle myopathy" to "myopathy" (page 19, line 431).

References

- 1 Walsh, S., Liu, D., Metter, E. J., Ferrucci, L. & Roth, S. M. ACTN3 genotype is associated with muscle phenotypes in women across the adult age span. *Journal of applied physiology* **105**, 1486-1491, doi:10.1152/jappphysiol.90856.2008 (2008).
- 2 Cho, J., Lee, I. & Kang, H. ACTN3 Gene and Susceptibility to Sarcopenia and Osteoporotic Status in Older Korean Adults. *Biomed Res Int* **2017**, 4239648, doi:10.1155/2017/4239648 (2017).
- 3 Judson, R. N. *et al.* The functional ACTN3 577X variant increases the risk of falling in older females: results from two large independent cohort studies. *J Gerontol A Biol Sci Med Sci* **66**, 130-135, doi:10.1093/gerona/glq189 (2011).
- 4 Ha, Y. C., Won Won, C., Kim, M., Chun, K. J. & Yoo, J. I. SARC-F as a Useful Tool for Screening Sarcopenia in Elderly Patients with Hip Fractures. *J Nutr Health Aging* **24**, 78-82, doi:10.1007/s12603-019-1307-6 (2020).
- 5 Chen, L. K. *et al.* Asian Working Group for Sarcopenia: 2019 Consensus Update on Sarcopenia Diagnosis and Treatment. *J Am Med Dir Assoc* **21**, 300-307 e302, doi:10.1016/j.jamda.2019.12.012 (2020).
- 6 Cruz-Jentoft, A. J. *et al.* Sarcopenia: revised European consensus on definition and diagnosis. *Age Ageing* **48**, 16-31, doi:10.1093/ageing/afy169 (2019).
- 7 Ogawa, S., Yakabe, M. & Akishita, M. Age-related sarcopenia and its pathophysiological bases. *Inflamm Regen* **36**, 17, doi:10.1186/s41232-016-0022-5 (2016).
- 8 North, K. N. *et al.* A common nonsense mutation results in alpha-actinin-3 deficiency in the general population. *Nature genetics* **21**, 353-354, doi:10.1038/7675 (1999).
- 9 MacArthur, D. G. & North, K. N. A gene for speed? The evolution and function of alpha-actinin-3. *Bioessays* **26**, 786-795, doi:10.1002/bies.20061 (2004).
- 10 Garatachea, N. *et al.* ACTN3 R577X polymorphism and explosive leg-muscle power in elite basketball players. *Int J Sports Physiol Perform* **9**, 226-232, doi:10.1123/ijsp.2012-0331 (2014).
- 11 San Juan, A. F. *et al.* Does complete deficiency of muscle alpha actinin 3 alter functional capacity in elderly women? A preliminary report. *Br J Sports Med* **40**, e1, doi:10.1136/bjism.2005.019539 (2006).

- 12 Delmonico, M. J. *et al.* Association of the ACTN3 genotype and physical functioning with age in older adults. *J Gerontol A Biol Sci Med Sci* **63**, 1227-1234, doi:10.1093/gerona/63.11.1227 (2008).
- 13 McCauley, T., Mastana, S. S. & Folland, J. P. ACE I/D and ACTN3 R/X polymorphisms and muscle function and muscularity of older Caucasian men. *Eur J Appl Physiol* **109**, 269-277, doi:10.1007/s00421-009-1340-y (2010).
- 14 Eynon, N. *et al.* The ACTN3 R577X polymorphism across three groups of elite male European athletes. *PLoS One* **7**, e43132, doi:10.1371/journal.pone.0043132 (2012).
- 15 Talbot, J. & Maves, L. Skeletal muscle fiber type: using insights from muscle developmental biology to dissect targets for susceptibility and resistance to muscle disease. *Wiley Interdiscip Rev Dev Biol* **5**, 518-534, doi:10.1002/wdev.230 (2016).
- 16 Cesari, M., Landi, F., Vellas, B., Bernabei, R. & Marzetti, E. Sarcopenia and physical frailty: two sides of the same coin. *Front Aging Neurosci* **6**, 192, doi:10.3389/fnagi.2014.00192 (2014).
- 17 Seto, J. T. *et al.* ACTN3 genotype influences muscle performance through the regulation of calcineurin signaling. *J Clin Invest* **123**, 4255-4263, doi:10.1172/JCI67691 (2013).
- 18 MacArthur, D. G. *et al.* An Actn3 knockout mouse provides mechanistic insights into the association between alpha-actinin-3 deficiency and human athletic performance. *Hum Mol Genet* **17**, 1076-1086, doi:10.1093/hmg/ddm380 (2008).
- 19 Hogarth, M. W. *et al.* Analysis of the ACTN3 heterozygous genotype suggests that alpha-actinin-3 controls sarcomeric composition and muscle function in a dose-dependent fashion. *Hum Mol Genet* **25**, 866-877, doi:10.1093/hmg/ddv613 (2016).
- 20 Lee, F. X., Houweling, P. J., North, K. N. & Quinlan, K. G. How does alpha-actinin-3 deficiency alter muscle function? Mechanistic insights into ACTN3, the 'gene for speed'. *Biochim Biophys Acta* **1863**, 686-693, doi:10.1016/j.bbamcr.2016.01.013 (2016).
- 21 Garton, F. C. *et al.* The Effect of ACTN3 Gene Doping on Skeletal Muscle Performance. *Am J Hum Genet* **102**, 845-857, doi:10.1016/j.ajhg.2018.03.009 (2018).
- 22 Luther, P. K. The vertebrate muscle Z-disc: sarcomere anchor for structure and signalling. *J Muscle Res Cell Motil* **30**, 171-185, doi:10.1007/s10974-009-9189-6 (2009).
- 23 Frank, D., Kuhn, C., Katus, H. A. & Frey, N. The sarcomeric Z-disc: a nodal point in signalling and disease. *J Mol Med (Berl)* **84**, 446-468, doi:10.1007/s00109-005-0033-1 (2006).
- 24 Demontis, F., Piccirillo, R., Goldberg, A. L. & Perrimon, N. Mechanisms of skeletal muscle aging: insights from *Drosophila* and mammalian models. *Dis Model Mech* **6**, 1339-1352, doi:10.1242/dmm.012559 (2013).
- 25 Kedia, N. *et al.* Desmin forms toxic, seeding-competent amyloid aggregates that persist in muscle fibers. *Proc Natl Acad Sci U S A* **116**, 16835-16840, doi:10.1073/pnas.1908263116

- (2019).
- 26 Maerkens, A. *et al.* New insights into the protein aggregation pathology in myotilinopathy by combined proteomic and immunolocalization analyses. *Acta Neuropathol Commun* **4**, 8, doi:10.1186/s40478-016-0280-0 (2016).
- 27 Piec, I. *et al.* Differential proteome analysis of aging in rat skeletal muscle. *FASEB J* **19**, 1143-1145, doi:10.1096/fj.04-3084fje (2005).
- 28 Gautel, M. & Djinovic-Carugo, K. The sarcomeric cytoskeleton: from molecules to motion. *J Exp Biol* **219**, 135-145, doi:10.1242/jeb.124941 (2016).
- 29 Haubold, K. *et al.* Acute effects of desmin mutations on cytoskeletal and cellular integrity in cardiac myocytes. *Cell Motil Cytoskeleton* **54**, 105-121, doi:10.1002/cm.10090 (2003).
- 30 Tahir, U., Monroy, J. A., Rice, N. A. & Nishikawa, K. C. Effects of a titin mutation on force enhancement and force depression in mouse soleus muscles. *J Exp Biol* **223**, doi:10.1242/jeb.197038 (2020).
- 31 Grison, M., Merkel, U., Kostan, J., Djinovic-Carugo, K. & Rief, M. alpha-Actinin/titin interaction: A dynamic and mechanically stable cluster of bonds in the muscle Z-disk. *Proc Natl Acad Sci U S A* **114**, 1015-1020, doi:10.1073/pnas.1612681114 (2017).
- 32 Roostaeae, A., Beaudoin, S., Staskevicius, A. & Roucou, X. Aggregation and neurotoxicity of recombinant alpha-synuclein aggregates initiated by dimerization. *Mol Neurodegener* **8**, 5, doi:10.1186/1750-1326-8-5 (2013).
- 33 Kim, D. *et al.* Identification of disulfide cross-linked tau dimer responsible for tau propagation. *Sci Rep* **5**, 15231, doi:10.1038/srep15231 (2015).
- 34 Kim, J. *et al.* Dimerization, oligomerization, and aggregation of human amyotrophic lateral sclerosis copper/zinc superoxide dismutase 1 protein mutant forms in live cells. *J Biol Chem* **289**, 15094-15103, doi:10.1074/jbc.M113.542613 (2014).
- 35 Roostaeae, A., Cote, S. & Roucou, X. Aggregation and amyloid fibril formation induced by chemical dimerization of recombinant prion protein in physiological-like conditions. *J Biol Chem* **284**, 30907-30916, doi:10.1074/jbc.M109.057950 (2009).
- 36 Sandri, M. & Robbins, J. Proteotoxicity: an underappreciated pathology in cardiac disease. *J Mol Cell Cardiol* **71**, 3-10, doi:10.1016/j.yjmcc.2013.12.015 (2014).
- 37 Brancolini, C. & Iuliano, L. Proteotoxic Stress and Cell Death in Cancer Cells. *Cancers (Basel)* **12**, doi:10.3390/cancers12092385 (2020).
- 38 Fernando, R., Drescher, C., Nowotny, K., Grune, T. & Castro, J. P. Impaired proteostasis during skeletal muscle aging. *Free Radic Biol Med* **132**, 58-66, doi:10.1016/j.freeradbiomed.2018.08.037 (2019).
- 39 Salomons, F. A. *et al.* Selective accumulation of aggregation-prone proteasome substrates in response to proteotoxic stress. *Mol Cell Biol* **29**, 1774-1785, doi:10.1128/MCB.01485-08 (2009).

- 40 Smith, L. R. & Barton, E. R. SMASH - semi-automatic muscle analysis using segmentation of histology: a MATLAB application. *Skelet Muscle* **4**, 21, doi:10.1186/2044-5040-4-21 (2014).
- 41 Dunant, P. *et al.* Expression of dystrophin driven by the 1.35-kb MCK promoter ameliorates muscular dystrophy in fast, but not in slow muscles of transgenic mdx mice. *Mol Ther* **8**, 80-89, doi:10.1016/s1525-0016(03)00129-1 (2003).
- 42 Liu, N. *et al.* Mice lacking microRNA 133a develop dynamin 2-dependent centronuclear myopathy. *J Clin Invest* **121**, 3258-3268, doi:10.1172/JCI46267 (2011).
- 43 Bruning, J. C. *et al.* A muscle-specific insulin receptor knockout exhibits features of the metabolic syndrome of NIDDM without altering glucose tolerance. *Mol Cell* **2**, 559-569, doi:10.1016/s1097-2765(00)80155-0 (1998).
- 44 Tai, P. W. *et al.* Differentiation and fiber type-specific activity of a muscle creatine kinase intronic enhancer. *Skelet Muscle* **1**, 25, doi:10.1186/2044-5040-1-25 (2011).
- 45 Miller, T. M. *et al.* Gene transfer demonstrates that muscle is not a primary target for non-cell-autonomous toxicity in familial amyotrophic lateral sclerosis. *Proc Natl Acad Sci U S A* **103**, 19546-19551, doi:10.1073/pnas.0609411103 (2006).
- 46 Cohn, R. D. *et al.* Disruption of DAG1 in differentiated skeletal muscle reveals a role for dystroglycan in muscle regeneration. *Cell* **110**, 639-648, doi:10.1016/s0092-8674(02)00907-8 (2002).
- 47 Beedle, A. M. *et al.* Mouse fukutin deletion impairs dystroglycan processing and recapitulates muscular dystrophy. *J Clin Invest* **122**, 3330-3342, doi:10.1172/JCI63004 (2012).
- 48 Kim, J. H. *et al.* Sex hormones establish a reserve pool of adult muscle stem cells. *Nat Cell Biol* **18**, 930-940, doi:10.1038/ncb3401 (2016).
- 49 Yin, H., Price, F. & Rudnicki, M. A. Satellite cells and the muscle stem cell niche. *Physiol Rev* **93**, 23-67, doi:10.1152/physrev.00043.2011 (2013).
- 50 Sambasivan, R. *et al.* Pax7-expressing satellite cells are indispensable for adult skeletal muscle regeneration. *Development* **138**, 3647-3656, doi:10.1242/dev.067587 (2011).
- 51 Relaix, F. & Zammit, P. S. Satellite cells are essential for skeletal muscle regeneration: the cell on the edge returns centre stage. *Development* **139**, 2845-2856, doi:10.1242/dev.069088 (2012).
- 52 Fry, C. S. *et al.* Inducible depletion of satellite cells in adult, sedentary mice impairs muscle regenerative capacity without affecting sarcopenia. *Nature medicine* **21**, 76-80, doi:10.1038/nm.3710 (2015).
- 53 Sousa-Victor, P. *et al.* Geriatric muscle stem cells switch reversible quiescence into senescence. *Nature* **506**, 316-321, doi:10.1038/nature13013 (2014).
- 54 Garcia-Prat, L. *et al.* Autophagy maintains stemness by preventing senescence. *Nature* **529**,

- 37-42, doi:10.1038/nature16187 (2016).
- 55 Finch, C. E., Felicio, L. S., Mobbs, C. V. & Nelson, J. F. Ovarian and steroidal influences on neuroendocrine aging processes in female rodents. *Endocr Rev* **5**, 467-497, doi:10.1210/edrv-5-4-467 (1984).
- 56 Nelson, J. F., Karelus, K., Bergman, M. D. & Felicio, L. S. Neuroendocrine involvement in aging: evidence from studies of reproductive aging and caloric restriction. *Neurobiol Aging* **16**, 837-843; discussion 855-836, doi:10.1016/0197-4580(95)00072-m (1995).
- 57 Cattanach, B. M., Iddon, C. A., Charlton, H. M., Chiappa, S. A. & Fink, G. Gonadotrophin-releasing hormone deficiency in a mutant mouse with hypogonadism. *Nature* **269**, 338-340, doi:10.1038/269338a0 (1977).
- 58 Koo, B. K. *et al.* Mind bomb 1 is essential for generating functional Notch ligands to activate Notch. *Development* **132**, 3459-3470, doi:10.1242/dev.01922 (2005).
- 59 Koo, B. K. *et al.* An obligatory role of mind bomb-1 in notch signaling of mammalian development. *PLoS One* **2**, e1221, doi:10.1371/journal.pone.0001221 (2007).
- 60 Romanick, M., Thompson, L. V. & Brown-Borg, H. M. Murine models of atrophy, cachexia, and sarcopenia in skeletal muscle. *Biochim Biophys Acta* **1832**, 1410-1420, doi:10.1016/j.bbadis.2013.03.011 (2013).
- 61 Nishikawa, T. *et al.* Tubular aggregates in the skeletal muscle of the senescence-accelerated mouse; SAM. *Mech Ageing Dev* **114**, 89-99, doi:10.1016/s0047-6374(00)00088-9 (2000).
- 62 Guo, A. Y. *et al.* Muscle mass, structural and functional investigations of senescence-accelerated mouse P8 (SAMP8). *Exp Anim* **64**, 425-433, doi:10.1538/expanim.15-0025 (2015).
- 63 Luo, J. *et al.* Nasal delivery of nerve growth factor rescue hypogonadism by up-regulating GnRH and testosterone in aging male mice. *EBioMedicine* **35**, 295-306, doi:10.1016/j.ebiom.2018.08.021 (2018).
- 64 Flood, J. F., Morley, P. M. & Morley, J. E. Age-related changes in learning, memory, and lipofuscin as a function of the percentage of SAMP8 genes. *Physiol Behav* **58**, 819-822, doi:10.1016/0031-9384(95)00125-3 (1995).
- 65 Ota, H. *et al.* Testosterone deficiency accelerates neuronal and vascular aging of SAMP8 mice: protective role of eNOS and SIRT1. *PLoS One* **7**, e29598, doi:10.1371/journal.pone.0029598 (2012).
- 66 Takeda, T. *et al.* A new murine model of accelerated senescence. *Mech Ageing Dev* **17**, 183-194, doi:10.1016/0047-6374(81)90084-1 (1981).
- 67 Zuriaga, M. A., Fuster, J. J., Gokce, N. & Walsh, K. Humans and Mice Display Opposing Patterns of "Browning" Gene Expression in Visceral and Subcutaneous White Adipose Tissue Depots. *Front Cardiovasc Med* **4**, 27, doi:10.3389/fcvm.2017.00027 (2017).

- 68 Rogers, N. H., Landa, A., Park, S. & Smith, R. G. Aging leads to a programmed loss of brown adipocytes in murine subcutaneous white adipose tissue. *Aging Cell* **11**, 1074-1083, doi:10.1111/accel.12010 (2012).

Reviewers' Comments:

Reviewer #1:

Remarks to the Author:

The authors have done significant work to address reviewer concerns and it has improved the manuscript.

The only remaining comment I have is to double-check the statistical symbols on Figure 5n. These show significance between the "outer" CSA values, and according to the response document these should not be different.

Reviewer #2:

Remarks to the Author:

The authors satisfactorily addressed the criticisms raised in the first revision. Still I would like to briefly discuss a couple of points:

- the authors show in their reply (point 7) that the mRNA expression of muscle-specific ubiquitin ligases is unchanged in Mib1 Δ MF mice in comparison to wild type animals, suggesting that muscle atrophy is not achieved by enhanced protein breakdown, at least due to the ubiquitin-proteasome proteolytic system. Can they provide data or discuss about the involvement of autophagic protein degradation? Do they believe that muscle atrophy results from reduced protein synthesis rates only?
- can the authors better define what do they mean with loss of myofibers by apoptosis?

Point-by-point reply in response to the reviewer's comments

Reviewer 1

The authors have done significant work to address reviewer concerns and it has improved the manuscript.

1. The only remaining comment I have is to double-check the statistical symbols on Figure 5n. These show significance between the "outer" CSA values, and according to the response document these should not be different.

- We apologize for the confusion in Figure 5n. As the reviewer pointed out, there is no significance between CSA of shCtrl^{outer} and shActn3^{outer}. We have corrected the statistical symbols on Figure 5n.

New Figure 5n. CSA of outer and inner part of Δ MF-EX+shCtrl and Δ MF-EX+shActn3.

Reviewer 2

The authors satisfactorily addressed the criticisms raised in the first revision. Still I would like to briefly discuss a couple of points:

1. The authors show in their reply (point 7) that the mRNA expression of muscle-specific ubiquitin ligases is unchanged in Mib1 Δ MF mice in comparison to wild type animals, suggesting that muscle atrophy is not achieved by enhanced protein breakdown, at least due to the ubiquitin-proteasome proteolytic system. Can they provide data or discuss about the involvement of autophagic protein degradation? Do they believe that muscle atrophy results from reduced protein synthesis rates only?

- Although there were no major changes in the expression levels of muscle-specific E3 ubiquitin ligases (MuRF1 and Atrogin-1) in Mib1 Δ MF mice (Extended Data Fig. 3e),

it doesn't imply that 'muscle atrophy is not achieved by enhanced protein breakdown, at least due to the ubiquitin-proteasome proteolytic system' as the reviewer pointed out. Although MuRF1 and Atrogin-1 are well-known E3 ubiquitin ligases implicated in skeletal muscle atrophy, they cannot account for the degradation of all sarcomeric proteins¹. Moreover, Lodka et al. showed myosin accumulation, and increased proteolysis and ubiquitin-proteasome proteolytic system in MuRF2 and MuRF3 double knockout skeletal muscle via proteomic analysis, suggesting that proteolytic pathways, specifically ubiquitin-proteasome proteolytic pathway can be activated to compensate for the loss of MuRF2 and MuRF3². Likewise, it is difficult to exclude the implication of ubiquitin-proteasome proteolytic system just because of comparable expression levels of Atrogin-1 and MuRF1 in Mib1^{ΔMF} mice.

During muscle atrophy, other proteolytic pathways such as autophagy system can be involved in the degradation of proteins, organelles, and protein aggregates³. Moreover, the sustained accumulation of protein aggregates can overburden autophagy-lysosomal activity with increased autophagosome synthesis accompanying excessive accumulation or production of autophagosome. Eventually, it can result in compromised autophagy-lysosomal activity and further contribute to proteotoxicity^{4,5}. Interestingly, according to the prediction algorithm Waltz, which predicts the amyloid regions⁶, there are 2 highly amyloidogenic regions, from aa 139 to 153 and from aa 688 to 699, in Actn3, indicating Actn3 can form either amorphous aggregates or amyloid fibrils. In that, it is possible that the sustained accumulation of Actn3 aggregates can induce excessive autophagy and overburden the autophagic capacity of Mib1^{ΔMF} mice.

As discussed above, we don't believe that muscle atrophy of Mib1^{ΔMF} mice is due to reduced protein synthesis rates only since proteolytic pathways can be enhanced to compensate for loss of E3 ubiquitin ligases.

Collectively, we added the detailed description in the discussion section (page 18, line 408-411; 418-431 and page 19, line 449-470).

2. Can the authors better define what do they mean with loss of myofibers by apoptosis?

- During atrophic conditions, myofiber can undergo programmed cell death induced by various cellular stresses such as proteotoxic stress, oxidative stress, nutritional stress, or protein misfolding, resulting in loss of myofibers^{3,7,8}.

Collectively, we added the detailed description in the discussion section (page 17, line 397-399; page 18, line 408-411; 418-431 and page 19, line 449-470).

References

- 1 Hnia, K., Clausen, T. & Moog-Lutz, C. Shaping Striated Muscles with Ubiquitin Proteasome System in Health and Disease. *Trends Mol Med* **25**, 760-774, doi:10.1016/j.molmed.2019.05.008 (2019).
- 2 Lodka, D. *et al.* Muscle RING-finger 2 and 3 maintain striated-muscle structure and function. *J Cachexia Sarcopenia Muscle* **7**, 165-180, doi:10.1002/jcsm.12057 (2016).
- 3 Fernando, R., Drescher, C., Nowotny, K., Grune, T. & Castro, J. P. Impaired proteostasis during skeletal muscle aging. *Free Radic Biol Med* **132**, 58-66, doi:10.1016/j.freeradbiomed.2018.08.037 (2019).
- 4 Button, R. W., Roberts, S. L., Willis, T. L., Hanemann, C. O. & Luo, S. Accumulation of autophagosomes confers cytotoxicity. *J Biol Chem* **292**, 13599-13614, doi:10.1074/jbc.M117.782276 (2017).
- 5 Jaeger, P. A. & Wyss-Coray, T. All-you-can-eat: autophagy in neurodegeneration and neuroprotection. *Mol Neurodegener* **4**, 16, doi:10.1186/1750-1326-4-16 (2009).
- 6 Maurer-Stroh, S. *et al.* Exploring the sequence determinants of amyloid structure using position-specific scoring matrices. *Nat Methods* **7**, 237-242, doi:10.1038/nmeth.1432 (2010).
- 7 Cohen, S., Nathan, J. A. & Goldberg, A. L. Muscle wasting in disease: molecular mechanisms and promising therapies. *Nat Rev Drug Discov* **14**, 58-74, doi:10.1038/nrd4467 (2015).
- 8 Cheema, N., Herbst, A., McKenzie, D. & Aiken, J. M. Apoptosis and necrosis mediate skeletal muscle fiber loss in age-induced mitochondrial enzymatic abnormalities. *Aging Cell* **14**, 1085-1093, doi:10.1111/accel.12399 (2015).